# Vapor-phased fabrication and modulation of cell-laden scaffolding materials

Chih-Yu Wu[1,2,3,7], Ting-Ying Wu[1,7], Zhen-Yu Guan[1], Peng-Yuan Wang[4], Yen-Ching Yang[1], Chao-Wei Huang[5], Tzu-Hung Lin[6] & Hsien-Yeh Chen [1,2,3 ✉]

Bottom–up approaches using building blocks of modules to fabricate scaffolds for tissue engineering applications have enabled the fabrication of structurally complex and multi-functional materials allowing for physical and chemical flexibility to better mimic the native extracellular matrix. Here we report a vapor-phased fabrication process for constructing three-dimensional modulated scaffold materials via simple steps based on controlling mass transport of vapor sublimation and deposition. We demonstrate the fabrication of scaffolds comprised of multiple biomolecules and living cells with built-in boundaries separating the distinct compartments containing defined biological configurations and functions. We show that the fabricated scaffolds have mass production potential. We demonstrate overall >80% cell viability of encapsulated cells and that modulated scaffolds exhibit enhanced cell proliferation, osteogenesis, and neurogenesis, which can be assembled into various geometric configurations. We perform cell co-culture experiments to show independent osteogenesis and angiogenesis activities from separate compartments in one scaffold construct.

[1] Department of Chemical Engineering, National Taiwan University, Taipei, Taiwan. [2] Molecular Imaging Center, National Taiwan University, Taipei, Taiwan. [3] Advanced Research Center for Green Materials Science and Technology, National Taiwan University, Taipei, Taiwan. [4] Center for Human Tissues and Organs Degeneration, Institute of Biomedicine and Biotechnology, Shenzhen Institutes of Advanced Technology, Chinese Academy of Sciences, Shenzhen, China. [5] Department of Tropical Agriculture and International Cooperation, National Pingtung University of Science and Technology, Pingtung, Taiwan. [6] Material and Chemical Research Laboratories, Industrial Technology Research Institute, Hsinchu, Taiwan. [7] These authors contributed equally: Chih-Yu Wu and Ting-Ying Wu. ✉email: hsychen@ntu.edu.tw

Modular approaches are used to produce scaffolding materials for tissue engineering that have delivered promising results for bottom–up methods, which show superior advantages, including more complex molecular and structural flexibility allowing for multifunctional assembly both in physical and chemical aspects to better mimic the native extra-cellular matrix (ECM)[1–3]. Current methods to apply module assemblies include direct assembly and/or aggregation, cell-laden hydrogels, cell sheets, or direct 2D or 3D printing techniques and are reported and reviewed elsewhere[3–10]. Questions ranging from cell–cell aggregation, distribution and localization of multiple types of cells, co-culture of cell types, distribution and localization of biomolecules, limited and potential immune-responsive materials, lack of surface modification of materials to the control of cell–material interactions remain unanswered. Answering these questions requires full integration of the cells and the surrounding biomolecules into the modulated materials. For instance, a key feature is the control of the distribution and location of cell-laden and biomolecular functionality within one module, which is important compared to post-seeding and functionalization after fabrication[11]. Extra caution is required to handle fabrication conditions, such as minimizing the involved solvents, chemicals, and high energy, to ensure viability, identity, and functionality of the targeted cells within the constructed scaffolds[12]. With these stringent fabrication requirements, only sporadic methods including cytocompatible hydrogels, bioinks for printing or spinning from 2D to 3D, and/or similar approaches are available thus far. The challenges, however, are that hydrogels provide excellent cellular cytocompatibility owing to their hydrated material properties but usually lack overall high mechanical integrity[12,13], and exist potential toxic degradation products;[14,15] the bioinks printing techniques, on the other hand, although provide excellent automated hierarchical control and with high shape fidelity, and are formulated to accommodate a wide range of applications, sensitive molecules, and cell suffers irritations by shear stresses during extrusion and injection[16,17], by additional energy sources or chemicals are involved, special modulation equipment is required for the fabrication[17–21], and unavoidable phase separation and dislocation of biomolecules and/or laden cells occurs owing to mass transport in such liquid-phased bioinks[22]. Ideal modular fabrication has yet to be achieved[6], and its combination with biological complexity may pave the way for the development of advanced scaffolds.

In this report, three-dimensional (3D) bulk polymeric materials with good control over shape and size at various scales are built from sublimating ice templates through a vapor-phase construction mechanism[23,24], which included the introduction of dopants in the ice templates, resulting in hierarchical internal porous structures and localized compartmentalization of the dopants. With using volatile compounds as dopants, the mechanical properties of the resultant porous materials are tuneable in a wide range, for instance, pore sizes ranging from >5 μm to 100 μm, and porosities of ~50–80%, Young's modulus from ~10 kPa to 10,000 kPa, are attempted and tuned;[25] while the use of non-volatile dopants render a uniform or controlled localization of encapsulating the dopants and without interference to the mechanical properties[23,26]. We additionally hypothesize that instead of sacrificing the dopants, we could utilize the functionalization substituents as preloaded components to include functional biomolecules and even living cells to build a cell-laden, modulated, multifunctional scaffold (Fig. 1a). The fabricated scaffold modules offer the advantages of (I) a straightforward accommodation of chemical/biological composition with multiple components ranging from functional biomolecules to living cells with determined composition ratio and customizable combination of these components, (II) a benign vapor-phase fabrication process, which utilizes ice/water templates for vapor sublimation and a USP (United States Pharmacopeia) Class VI highly bio-compatible poly-p-xylylene for vapor deposition, forming a scaffold matrix without irritating sensitive molecules and cells, (III) control mass transport of species capable of being operated at an unsteady-state and/or steady-state conditions in a defined construction volume to avoid phase separation and dislocation of the components, (IV) connect pore structure formation with tuneable mechanical properties[25] allowing for interaction between the preloaded components and cells, and (V) a robust discontinued process of assembling modules during the templating stage, followed by a continued, one-step vapor deposition process resulting in a continuous scaffold construct composed of the spatial arrangements of specified functional modules with established boundaries between different cell types and micro-environments. The scaffold is designed to provide not only passive structural support but also biological cues to the inhabitant cells for guided cell attachment, proliferation, and differentiation.

## Results and discussion

**Vapor-phased scaffold fabrication and modulation.** The fabrication process was initiated by preparing an iced template from a solution with predetermined solute compositions rather than using sacrificing dopants as in the previous reports[25]. Live cells, including hASCs (human adipose-derived stem cells), MC3T3-E1 (mouse pre-osteoblastic cell line), PC12 (rat adrenal phaeochromocytoma cell line), HUVECs (human umbilical vein endothelial cells), and MG-63 (human osteosarcoma cell line) as well as the growth factor proteins BMP-2 (bone morphogenetic protein), FGF-2 (fibroblast growth factor), VEGF (vascular endothelial growth factor), and PRP (platelet-rich plasma) were used to prepare the solution for subsequent solidification to form iced templates. Specifically, a module containing FGF-2 and hASCs was fabricated, which was used to demonstrate the cell proliferation, was first selected for preparing the iced templated. Subsequently, based on the unique vapor sublimation and deposition mechanism[23,27,28], the iced template was sublimated with the evaporation of water molecules, and a depositing poly-p-xylylene polymer occurred to replace the evaporated volume of water and forming a porous scaffold containing the preloaded FGF-2 and hASCs. During the fabrication process, an analysis by using a mass spectrometric gas analyser (RGA) verified the sublimated water molecules (18 amu) and the deposited quinodimethane derivatives (104 amu and 139 amu), as shown in Fig. 1b.

Similar to dopants, which are prepared in water solutions with proper mixing procedures, the cells, and biomolecules (FGF-2) were suspended in buffer solutions to obtain (i) cells in an oil-in-water subsystem and (ii) FGF-2 in the buffer solvent phase[25]. This arranged distribution of (i) and (ii) was then subjected to an instant solidification procedure by liquid nitrogen to form the iced template and to prevent a potential unsteady-state mass transport and an undesired result of mixing of (i) and (ii). After the iced template was prepared, the second and final fabrication step was the sublimation and deposition of poly-p-xylylene to obtain a final porous scaffold consisting of the poly-p-xylylene porous matrix with preloaded hASCs cells and FGF-2 biomolecules in the same arranged locations. The fabrication is a time-dependant process to produce a proportional volume of scaffold product and required ~60 mins for a 5 cm³-sized sample, and theoretically, size of the fabricated scaffold module is limited to the ice templates that can be produced by existing techniques, and can be tuned based on a stage-wise control and timing of the involved sublimation and deposition processes[23]. With also tuneable mechanical properties to fabricate the scaffold modules[25], consistent properties including ~35.7 ± 8.2 μm in pore

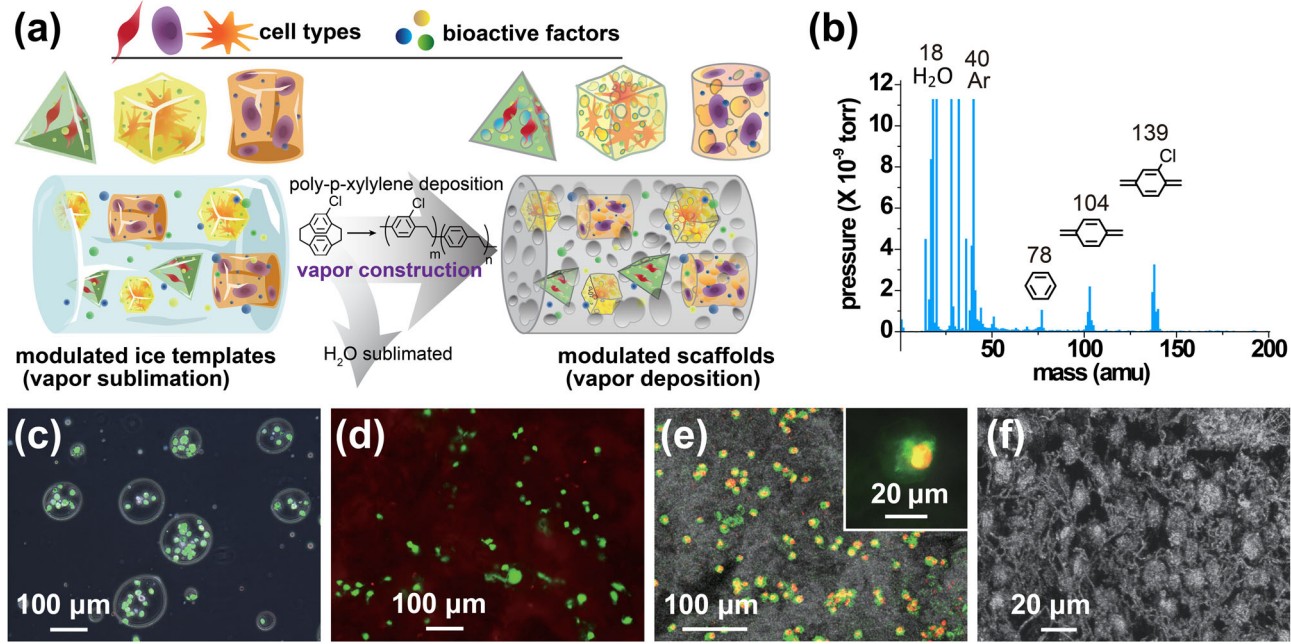

**Fig. 1 Fabrication of cell-laden and multifunctional scaffold modules. a** Schematic illustration of vapor-phase sublimation and deposition fabrication process. Iced template modules with defined internal composition and outer architecture were formed through discrete module assembly and resulted in a continuous final modulated construct by subsequent vapor deposition. **b** The analysis of vapor composition by real-time mass spectrometry during the fabrication process shows the presence of sublimated water vapor at 18 amu and the deposited quinodimethane monomers at 104 amu and 139 amu. **c** An overlaid image of LIVE/DEAD-stained hASCs prepared in an oil-in-water system. Verification of hASCs within a fabricated module after vapor-phase deposition process by **d** LIVE/DEAD staining, **e** fluorescence labeling with Alexa Fluor® 488-conjugated phalloidin to stain the cytoskeleton (green) and propidium iodide to stain the nucleus (red); a high-resolution and magnified image of a stained cell was shown in the inset, and **f** SEM dissection image. Each experiment was repeated three times independently ($n = 3$) with similar results.

size, 63.4% ± 6.3 porosity, and 150 ± 21.5 kPa for Young's modulus, were measured and used in the current studies. To ensure cell viability for cell-laden samples, immersing the samples in the culture medium was performed immediately upon retrieving the samples from the deposition chamber. The localized cells in the prepared oil-in-water system in the buffer solution are shown in Fig. 1c; these cells were confirmed to be alive after the vapor deposition process in the fabricated modules by combined characterizations using LIVE/DEAD staining, fluorescence labeling, and images captured by scanning electron microscopy (SEM), as shown in Fig. 1d–f. Removal of the oil from the scaffold system was expected based on previously discovered results[25]. An overall >80% rate of viable cells was estimated based on comparing the LIVE/DEAD signals, including 98.2% of viable cells in oil-in-water suspension, 96.1% after freezing in the ice templates, and 80.8% after fabrication in the final scaffold constructs. Additional data for analyzing the cell viability are also included in Supplementary Figure 1. The cell viabilities and compatibilities are expected extendable to various types of cell and stem cell systems by using the fabrication process and the poly-*p*-xylylene (and the derivatives) scaffold materials based on the compatibilities studies reported elsewhere[29–32].

**Modulations of biological functions and cell-guiding activities**. The important question was whether the modulated and arranged scaffolding functions can be implemented with cell viability, attachment of growing cells, and guided cell proliferation with respect to FGF-2. The answers to these questions were verified by culturing the preloaded scaffolds under the appropriate conditions. As revealed in Fig. 2a, the induced proliferation of hASCs in terms of their growth ability was evaluated at days 1 and 4 and compared with a control experiment, in which the preloaded cells were cultured without the composited growth factor proteins.

SEM characterizations identified increasing clusters of cells at day 4, showing enhanced attaching, spreading, and growth of hASCs on the porous structures. In addition, propidium iodide was used to stain the cell nucleus and Alexa Fluor™ 488 phalloidin was used to stain the actin filaments, and fluorescence microscopy was used to characterize the growth activities of the stained hASCs within the FGF-2-composited polymer systems in terms of cell number and distribution. The acquired fluorescence images show comparable results in terms of cell growth with slight differences in the cell numbers, and well-distributed cells were discovered in all the studied sample groups; the results also indicated a homogeneous distribution and inclusion of the hASCs by the proposed fabrication method on day 1, but no proliferation activity was detected[33]. By contrast, the cell growth activities on day 4 showed the anticipated proliferation, and a greatly increased number of homogeneously spread cells was observed in the FGF-2 composited group compared with the group without FGF-2. Separate experiments and characterizations by using an MTT (3-(4,5-dimethylthiazol-2-yl)-2,5-diphenyltetrazolium bromide) assay additionally showed statistically significant and consistent results, which suggested that the FGF-2-decorated samples resulted in enhanced proliferation of hASCs. Additional control experiments were performed by Live/Dead staining showing the comparison of the FGF-2-decorated samples with samples without FGF-2 at day 1 and day 4, and the data are included in Supplementary Figure 2. A modulated scaffold for inducing osteogenesis activity was also prepared by combining MC3T3-E1 cells with (i) and BMP-2 with (ii) in the preparation of the iced templates used for the fabrication of the scaffold products. The results were revealed in Fig. 2b, showing the osteogenesis activities were evaluated to detect the early stage osteogenesis marker of alkaline phosphatase (ALP) expression at day 7 and the late-stage markers of osteocalcin (OCN) expression

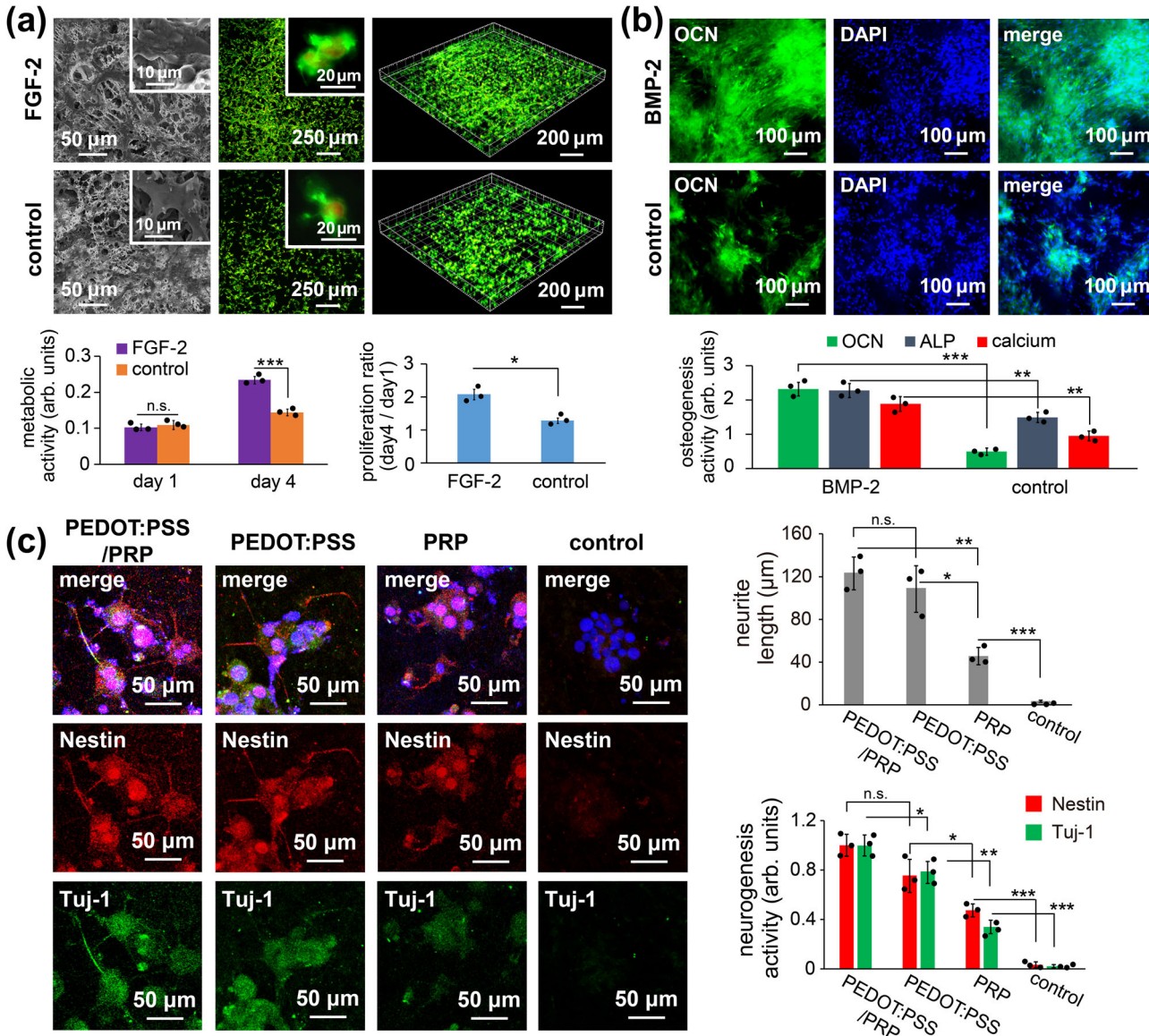

**Fig. 2 Fabrication of various modules with specified cell-guiding activities. a** SEM and laser confocal images were recorded for modules containing hASCs cells and FGF-2 and showed cell proliferation activities after culturing for 4 days. The F-actin cytoskeleton (green) was stained by Alexa Fluor® 488-conjugated phalloidin and the nucleus (red) was stained with propidium iodide; the inset images showed attached and spread cells ($n = 3$ independent samples; mean ± SD; unpaired $t$ test; bottom left panel: FGF-2 vs. control at day 1 n.s., nonsignificant difference $p = 0.7217 > 0.05$, FGF-2 vs. control at day 4 ***$p = 0.0006 < 0.001$; bottom right panel: FGF-2 vs. control *$p = 0.0116 < 0.05$). **b** Modules composed of MC3T3-E1 cells and BMP-2 showed guided osteogenesis activities. The ALP expression was analyzed after 7 days of culture, whereas OCN expression and calcium deposition were analyzed after 21 days of culture ($n = 3$ independent samples; mean ± SD; unpaired $t$ test; bottom panel: BMP-2 vs. control from OCN ***$p = 0.0002 < 0.001$, BMP-2 vs. control from ALP **$p = 0.0056 < 0.01$, BMP-2 vs. control from calcium **$p = 0.0033 < 0.01$). **c** Modules composed of more complex components, including PC12 cells, PRP molecules, and the PEDOT:PSS system, were fabricated and showed guided neurogenesis activities after 5 days of culture ($n = 3$ independent samples; mean ± SD; unpaired $t$ test; top panel: PEDOT:PSS/PRP vs. PEDOT:PSS n.s. $p = 0.3915 > 0.05$, PEDOT:PSS/PRP vs. PRP **$p = 0.0014 < 0.01$, PEDOT:PSS vs. PRP *$p = 0.0103 < 0.05$, PRP vs. control ***$p = 0.0009 < 0.001$; bottom panel: PEDOT:PSS/PRP vs. PEDOT:PSS from Nestin n.s. $p = 0.0617 > 0.05$, PEDOT:PSS/PRP vs. PEDOT:PSS from Tuj-1 *$p = 0.0452 < 0.05$, PEDOT:PSS vs. PRP from Nestin *$p = 0.0295 < 0.05$, PEDOT:PSS vs. PRP from Tuj-1 **$p = 0.0019 < 0.01$, PRP vs. control from Nestin ***$p = 0.0008 < 0.001$, PRP vs. control from Tuj-1 ***$p = 0.0008 < 0.001$).

and calcium mineralization at day 21. These results were compared with the results of the experimental groups with BMP-2 biomolecules decorated samples and the groups without BMP-2. Fluorescence images of OCN expression (green channel) unambiguously revealed evidence of osteogenesis promotion for the BMP-2-decorated samples. Statistically, the recorded intensities of the staining signals for OCN expression, ALP activity, and deposited calcium minerals were analyzed, and the results showed strong osteogenesis activities for the modulated groups, and

showed a significant increase (**$p$ value < 0.01) in osteogenesis when compared with pure scaffold systems with no BMP-2.

A more complicated scaffolding system for neurogenesis allowing for the addition of substances with orthogonal properties comprised (i) pheochromocytoma 12 cells (PC12), (ii) a multi-component guidance substance of PRP[34], and (iii) a newly introduced guidance substance of a conductive polymer system, poly(3,4-ethylenedioxythiophene) polystyrene sulfonate (PEDOT: PSS). The iced templates were prepared, and the corresponding

scaffold products comprising (i), (ii), and (iii) were constructed by the vapor process (theoretically, multiple and diverse solute entities besides the shown examples can be included in the modulation). Cell cultures with the resultant multifunctional module scaffolds (containing both (ii) and (iii)) were performed under programmed electrical stimulation, and their potential neurogenesis activities were evaluated and compared with module scaffold systems, which contained only a single functionality, i.e., solely (ii) or (iii), and with the pure scaffold containing no functionalities. The expression of two neural markers, Nestin and Tuj-1[35], indicative of neurogenesis activities, was then characterized by immunofluorescence staining using primary antibodies, Anti-Nestin antibody and Anti-beta III Tubulin antibody, and subsequently with fluorescence-conjugated secondary antibodies. The results in Fig. 2c revealed the presence of Nestin (red channel) and Tuj-1 (green channel) in the studied systems; the two markers showed distinguishable high intensities (10-fold more) in the studied system with the multifunctional entities of PRP and PEDOT:PSS in relation to the low intensities found in the control groups. Additional characterizations of the cell morphology were performed by staining the differentiated PC12 cells, and the resultant fluorescence images indicated significantly extended neurite structures of the cells in the multifunctional system, and the high-intensity markers Nestin and Tuj-1 were detected in the same system. The neurite length was calculated to be the longest $115.3 \pm 26.8\,\mu m$ in the multifunctional system, compared with $98.4 \pm 24.2\,\mu m$ and $45.6 \pm 8.1\,\mu m$ in the systems with a single functional entity, PRP or PEDOT:PSS, respectively; spherical cell aggregates without neurite formation were observed in the system without any functional entity, indicative of undifferentiated cells[36]. Additional data to characterize the inclusion of varied dopants materials in the fabricated scaffolds are included in Supplementary Figure 3.

**Geometric assembly of modules with defined chemical/biological boundaries**. The assembly of the module scaffolds into a spatial arrangement of interest and/or the assembly of these functional substances based on a combinatorial and complementary modulation to enable hierarchically complex cells and functional components as well was the establishment of the required boundaries between the different cell types and/or the compartmentalized microenvironments can result in a scaffold with spatial and cascade functional arrangement for tissue engineering. This concept was finally realized by assembling the prepared iced templates into modules, resulting in a single compartmentalized iced template, which combined the functional entities incorporated in each of the compartments. Discrete boundaries were established naturally between each assembled ice template/module, and the subsequent vapor sublimation of the ice template and simultaneous vapor deposition of polymeric poly-p-xylylene resulted in a continuous construct of a modulated scaffold product with the same established boundaries and compartments. A multifunctional, compartmentalized, and modulated scaffold system was obtained, which comprised the porous poly-p-xylylene polymer matrix with the same boundaries as the ice template established within the matrix separating two (or more) solute systems in the spatially distinct compartments. Various combinations of the two solution systems were prepared, and two iced template modules were formed for subsequent assembly into a single modulated scaffold constructed by the vapor-phase process; these assemblies could be obtained with characteristic shapes, which verified the modulation concept. In the demonstration, various combinations of two solution systems were prepared to form two iced template modules for the assembly, and the same vapor construction process transformed

the assembled iced templates to produce separate compartments of A and B with varied geometries and dimensions in one modulated scaffold. Demonstrations were shown with the versatility to configure compartment A vs. B in equal aspect ratio, asymmetrical, discontinued configuration, and curved distribution of these two compartments. For example, fluorescent solutions of Oil Red-O (red channel) vs. fluorescein-5-isothiocyanate (green channel) were used to generate a color contrast between the compartments; blank solution vs. solutions containing silver (Ag) nanoparticles were used to produce a modulated scaffold with high-density contrast between the compartments. These assemblies were characterized by laser confocal microscopy, as shown in Fig. 3a, and micro-computed tomography (micro-CT), as shown in Fig. 3b, which revealed varied shapes and dimensions implying the versatility of the assembly process. Defined boundaries, which differentiated the compartments based on density and color, were detected and corresponded to the shapes and dimensions. Calculated volume ratios for the studied configurations based on analyzing the 3D-profiled and micro-CT results were also obtained showing ~50% vs. 50% (equal aspect ratio), 56% vs. 44% (asymmetrical), 98% vs. 2% (discontinued configuration), and 26% vs. 74% (curved distribution) for compartment A vs. compartment B, respectively. A greater than 95% accuracy of the compartmentalized modules within one modulated scaffold construct was estimated, and the detailed dimensional data comparing the ice templates and the scaffold modules were summarized in Supplementary Table 1.

**Cell co-cultures of modulated scaffold constructs**. Most importantly, the ability to preload cells and biomolecules into the compartments of the iced templates and thus into the final modulated scaffolds was demonstrated as follows: compartment-(A) comprised human osteoblasts (MG-63) in (i) and BMP-2 in (ii), and compartment-(B) comprised HUVECs in (i) and VEGF in (ii). Specific staining procedures of each component were carefully performed in each compartment of the same modulated scaffold, as detailed in the Methods. The staining results are summarized in Supplementary Figure 4, and the overlaid images in Fig. 3c provided overview pictures showing the corresponding shapes and dimensions with defined module boundaries of the multiple cells and biomolecule compartments. Demonstrations of mass production potential to fabricate these modulated scaffolds in varied shapes and dimensions are also included in Supplementary Figure 5. With the versatility to produce a scaffold construct composed of the spatial arrangements of specified functional modules with established boundaries between different cell types and microenvironments that was demonstrated in Fig. 3c, more-detailed cell co-cultures were finally performed with this sophisticated module scaffold, and a demonstration of customizable and programmable biofunctionalities was arranged in the compartmentalized A and B with multiple cell types showing independent cascades of spatial and temporal guidance in such a modulated scaffold. Scaffolds with the distinct cell-laden compartments A and B were co-cultured under the same conditions and with the same culture media, and fabricated scaffold samples with the configuration of an equal aspect ratio for compartment A vs. compartment B was chosen for the demonstration. The experiments were performed based on the hypothesis of guided MG-63 cell differentiation towards osteogenesis through the BMP-2 molecular arrangement in compartment A; angiogenesis of HUVECs was achieved through arrangement of a second feeder cell, hASCs[37,38], and VEGF in compartment B. The combination was chosen in light of the recognized histological observation of human bone tissues, where osteoblasts and osteoprogenitor cells are located adjacent to blood vessel

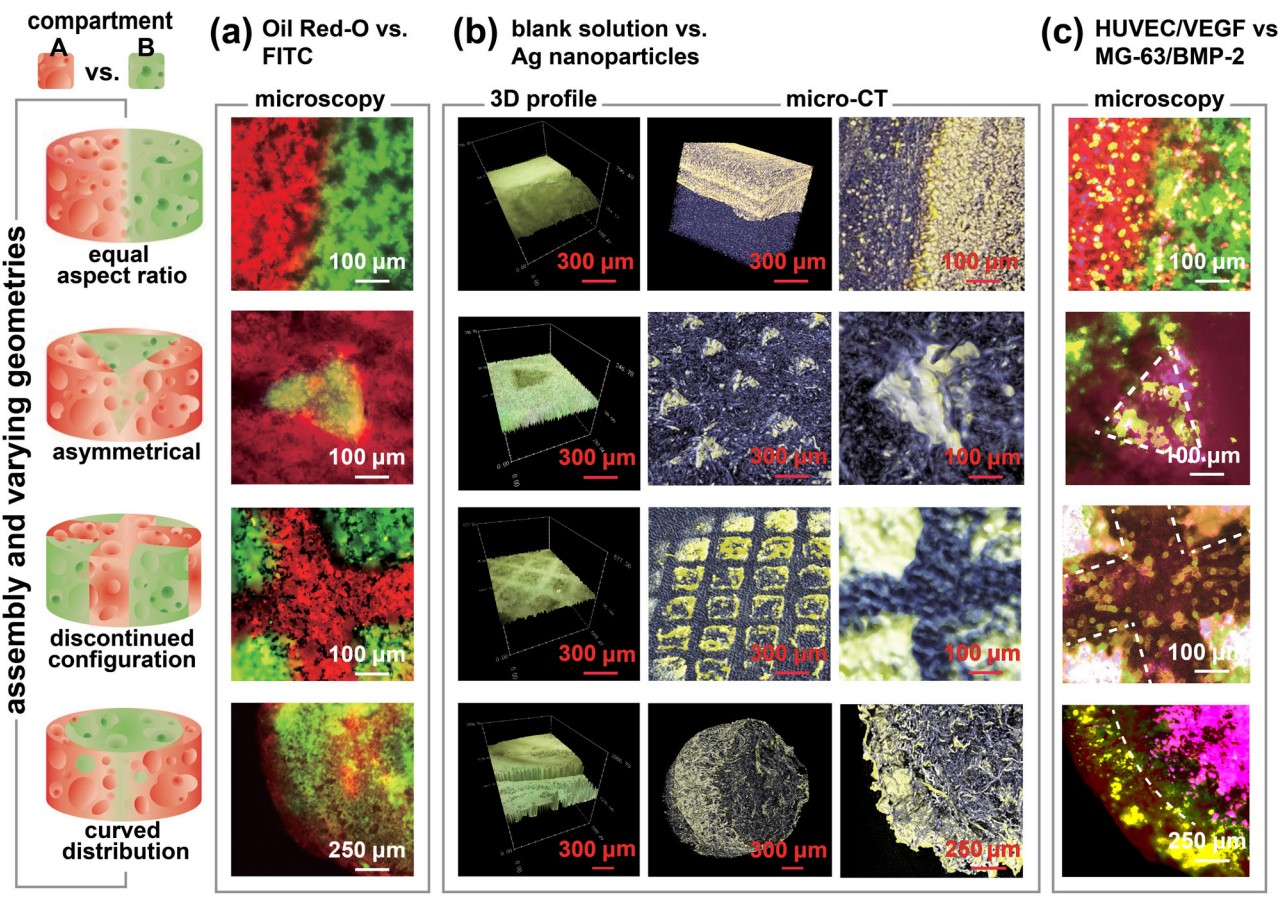

**Fig. 3 Assembly of various modules with defined chemical/biological components and geometric architecture into one modulated construct with different compartments A vs. B.** Components used for assembly and fabrication of the two compartments included **a** Oil Red-O (red channel) vs. fluorescein-5-isothiocyanate (green channel), **b** blank solution vs. silver (Ag, yellow) nanoparticles in 3D-profiled images and tomographic images, and **c** co-cultured system HUVECs combined with VEGF vs. MG-63 and BMP-2. The geometric illustration of the assembly and fabrication is shown on the left indicating equal aspect ratio, asymmetrical, discontinued configuration, and curved distribution of the two compartments. Each experiment was repeated three times independently ($n = 3$) with similar results.

endothelial cells[39,40]. The results are shown in Fig. 4a, and the cell morphologies were detected via immunofluorescence staining of the osteogenic marker collagen type-I (COL-I) and the angiogenic marker platelet endothelial cell adhesion molecule (PECAM-1/CD31); bone-forming cells originating from the preloaded MG-63 in compartment A and tubule-forming endothelial cells originating from the preloaded HUVECs in compartment B were discovered in the specified compartments. Additional data to reveal immunofluorescence characterizations and statistical analysis for confirmations of the specific MG-63 cells and HUVECs in compartment A and B are included in Supplementary Figure 6. Furthermore, the expression of both markers over time was evaluated based on the differentiation pathways of osteogenesis and angiogenesis;[41,42] the detected immunofluorescence signals showed an enhancement in COL-I expression of MG-63 cells from day 7 to day 21 corresponding to osteogenesis activity in compartment A and enhanced CD31 signals in vascularized HUVEC cells from day 3 to day 10 for potential angiogenesis activity in B. The independent and distinct biological activities detected in compartments A and B resulted from the cell differentiations in the corresponding compartments, which supported the hypothesis. Specifically, in compartment A, the formed bone-like nodules[43] were composed of COL-I-positive MG-63 cells, indicating vital osteogenic activity. At day 7, the cell density varied from 11 to 18 cells per nodule with an average size of 175 ± 22 μm. The size of the bone-like nodules increased during the

osteogenic differentiation, and 14-day-old nodules showed a size of 318 ± 69 μm with ~71 cells per nodule, whereas 21-day-old nodules had a size of 638 ± 65 μm with ~260 cells per nodule. On the other hand, in compartment B, the formed classic tube networks[37,38] were composed of CD31-positive HUVECs, indicating vital angiogenic activity. A clear network lumen was observed on day 7 and day 10, and the network parameters were measured, including total network length (sum of the lengths of all segments within the 3D network) and network branch points, which were 10.2 ± 1.0 mm and 30.1 ± 3.8 per area of interest (1 mm²), respectively, over 10 days of culture. Additional data indicated hASCs interacted HUVECs' network as a role of feeder cells during the HUVEC maturation by forming hollow lumen, and these data are included in Supplementary Figure 7. Statistical analysis of the osteogenic and angiogenic activities based on these recorded protein marker signals and morphologic changes were finally compared in the studied time frames of cell co-culture for both compartments, and the anticipated increase in their physiological activities was confirmed in Fig. 4b–c.

The results collectively and unambiguously verified the hypotheses that preloaded and inhabitant cells within the porous polymer system were viable, had the ability to proliferate, and retained their functions for differentiation, the compositing and fabrication process was flexible and versatile and allowed for the inclusion of functional entities, e.g., cells, growth factor proteins, and other molecules (without theoretical limitations), and the

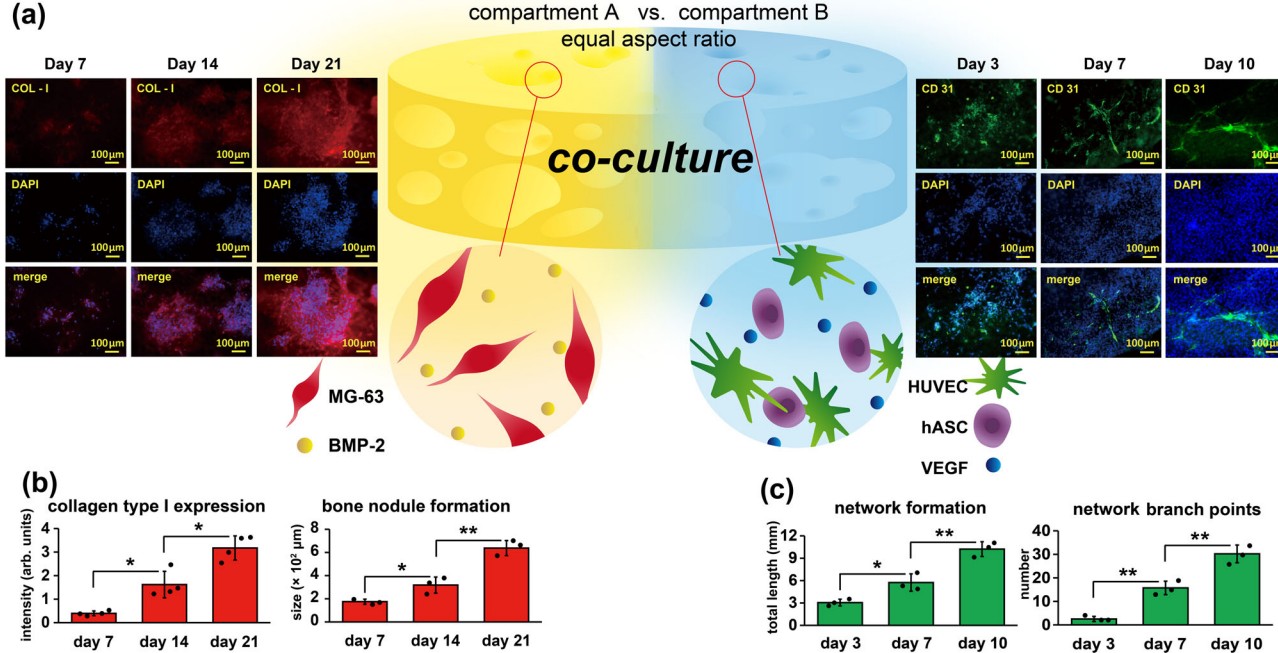

**Fig. 4 Cell co-cultures of the modulated constructs with defined and customized compartments of A containing the combination of MG-63 and BMP-2, and compartment B containing HUVECs, hASCs, and VEGF.** The fabricated scaffold samples with the configuration of an equal aspect ratio for compartment A vs. compartment B was chosen in the study. **a** The recorded immunofluorescence images of fibrous type-I collagen (COL-I) expression at day 7, day 14, and day 21 indicated guided osteogenic activities in compartment A during the co-cultures. The bone-like nodules formed and increased in size over the indicated period. On the other hand, representative immunofluorescence images of platelet endothelial cell adhesion molecule (PECAM-1/CD31) expression at day 3, day 7, and day 10 showed guided angiogenic activities in compartment B. The classic tube network formed and developed increasingly over the indicated period. **b** Quantitative analysis of osteogenic activities ($n = 4$ or 3 independent samples; mean ± SD; unpaired $t$ test; bottom left panel: day 7 vs. day 14 *$p = 0.0217 < 0.05$, day 14 vs. day 21 *$p = 0.0249 < 0.05$; bottom right panel: day 7 vs. day 14 *$p = 0.0268 < 0.05$, day 14 vs. day 21 **$p = 0.0043 < 0.01$) and **c** angiogenic activities of the MG-63 and HUVEC co-culture samples containing compartmentalized BMP-2 and VEGF in compartment A and compartment B, respectively ($n = 3$ independent samples; mean ± SD; unpaired $t$ test; bottom left panel: day 3 vs. day 7 *$p = 0.0202 < 0.05$, day 7 vs. day 10 **$p = 0.0071 < 0.01$; bottom right panel: day 3 vs. day 7 **$p = 0.0023 < 0.01$, day 7 vs. day 10 **$p = 0.0079 < 0.01$).

interacting functions between the included entities were orthogonal and multifunctional, and the osteogenesis and angiogenesis of the embedded cells were effectively guided by the fabricated 3D microenvironments of the porous polymer system.

This study took advantage of the complexity and multifunctionality of modulated building blocks to assemble scaffolds and provided sophisticated structural and biological guidance for future tissue engineering applications. The concept and fabrication method reported herein represented a sophisticated but simple process: chemically, the unique fabrication process employed dry and clean one-step vapor deposition to construct materials in the vapor phase and form a scaffold matrix consisting of polychloro-$p$-xylylenes (USP highest class VI) with excellent biocompatibility; physically, interconnected pores with high flexibility were formed to encapsulate multiple living cells and growth factors into the compartmentalized combinations of controlled geometric dimension and shape, and biologically, customizable and programmable biofunctionalities were arranged for the co-culture of multiple cell types showing cascades of spatial and temporal guidance. The design and application of modulated scaffold products are intuitively extendable beyond the examples described in the report, with unlimited applications for tissue engineering, cell-encapsulated drug delivery, regenerative treatments, among others.

## Methods

**Fabrication.** Iced templates were prepared from polydimethylsiloxane molds[44] of varied shapes and dimensions, including trigonal pyramids of 300 μm length, cubes of 300 μm length, and cylinders of 900 μm radius; the molding process was followed

by a solidification procedure under cooled conditions using liquid nitrogen. Solutions for preparing the iced templates ranged from 30 mg/ml silver nanoparticles (Sigma-Aldrich, USA), 40 vol% Oil Red-O (Sigma-Aldrich, USA), 0.5 mg/ml fluorescein-5-isothiocyanate (Sigma-Aldrich, USA), 10 ng/ml recombinant human FGF-2 (R&D Systems, USA), 100 ng/ml recombinant human BMP-2 (R&D Systems, USA), 50 ng/ml VEGF (Sigma-Aldrich, USA), PRP from a 7-day-old piglet (IACUC approval no. NTU-102-EL-3) by centrifugation[34] using a cell separator (Aeon Biotherapeutics, Taiwan), 5 wt% PEDOT:PSS (Sigma-Aldrich, USA), to specified cells in glyceryl trioleate (Sigma-Aldrich, USA) system (oil-in-water system)[25]. Assembly of the iced templates/modules was performed by blending solid-phased ice with liquid-phased solutions in specified compositions and in multiple cooling steps. The modulated ice templates were used for the deposition of polychloro-$p$-xylylene at 100 mTorr and 4 °C to undergo a simultaneous process of water vapor sublimation and $p$-xylene deposition[23,24] and form the final modulated scaffold product. A home-built sublimation/deposition system was used to perform this fabrication, which was composed of a pyrolysis furnace operated at 650 °C at the $p$-xylene inlet and a chamber with a temperature-controllable sample holder at 4 °C. The iced templates and the fabricated modulated scaffolds with varied shapes and sizes are shown in Supplementary Figure 3.

**Cell lines and culture conditions.** hASCs (ATCC® PCS-500-011™)[45], MC3T3-E1 mouse preosteoblast subclone (ATCC® CRL-2593™)[46], and PC12 cell line of rat adrenal pheochromocytoma cells (ATCC® CRL1721™)[47] were obtained from American Type Culture Collection (ATCC, USA). HUVECs (HBCRC No. H-UV001)[48] and human osteosarcoma/osteoblasts (MG-63, BCRC No. 60279)[49] were both obtained from the Bioresource Collection and Research Center (BCRC, Taiwan). hASCs, MC3T3-E1, and MG-63 cells were cultured in MEM-alpha (Thermo Fisher Scientific, USA) supplemented with 10% fetal bovine serum (FBS, Biological Industries, Israel) and 1% antibiotic/antimycotic solution (Biological Industries, Israel). PC12 cells were cultured in Roswell Park Memorial Institute (RPMI-1640) medium (Thermo Fisher Scientific, USA) supplemented with 5% FBS (Biological Industries, Israel), 10% heat-inactivated horse serum (Biological Industries, Israel), and 1% antibiotic/antimycotic solution (Biological Industries, Israel). HUVECs were cultured in M199 medium (Sigma-Aldrich, USA) supplemented with 25 U/ml heparin (Sigma-Aldrich, USA), 30 μg/ml endothelial cell growth supplement

(Sigma-Aldrich, USA), 10% FBS (Biological Industries, Israel), and 1% antibiotic/antimycotic solution (Biological Industries, Israel). Culturing conditions were maintained at 37 °C with a humidified atmosphere containing 5% $CO_2$ and 95% air, and media were changed once every 3–4 days. The cell loading density was $\sim 1 \times 10^5$ cells per sample during the experiments unless otherwise indicated, and the fabricated cell-laden scaffold samples were cultured immediately in the above-mentioned corresponding growth medium, or induction medium to identify osteogenesis and neurogenicity[50,51].

**Material characterization.** A real-time mass spectrometer (RGA, Hiden Analytical, UK) was used to monitor the sublimation and deposition, and the operating conditions were $10^{-7}$ mbar with a 70 eV electron ionization energy and emission current of 20 μA. The real-time mass spectrum was reconstructed by Hiden analytic software (MASsoft7 professional). The mass detection range was from 0 to 500 amu. SEM images were recorded with a Nova™ NanoSEM (FEI, USA) with primary energy of 10 keV and operating pressure of $5 \times 10^{-6}$ Torr and constructed by Nova NanoSEM software (version 1.3.1). FTIR spectra were recorded with a liquid nitrogen-cooled (MCT) Spectrum 100 FTIR Spectrometer (PerkinElmer, USA) equipped with an advanced grazing angle specular reflectance accessory (AGA, PIKE Technologies, USA) and collected by the Spectrum software (Version 6.3.5.0176). The scanning range was from 500 to 4000 cm$^{-1}$ with a 4 cm$^{-1}$ resolution with 128 scans. The three-dimensional (3D) microstructural and architectural features of the synthetic porous structures containing silver nanoparticles were assessed by micro-CT. The samples were scanned using a Skyscan 1272 high-resolution micro-CT (Bruker, Germany) with 40 kV, 250 μA, 10 W output, a pixel size of 2.5 μm and 4 K resolution. Ring artefacts and beam-hardening correction were performed with GPU-Nrecon software (version 1.7.1.0). Analysis was performed by using CTAn software (version 1.20.8). The 3D image visualization was performed by using CTVox (version 3.3.1).

**Analysis of cell behavior.** A fluorescence-based LIVE/DEAD kit (Thermo Fisher Scientific, USA) was used to characterize live or dead cells present on the studied samples after incubation for an indicated period of time. A commercial MTT assay kit (Sigma-Aldrich, USA) was used, and the absorbance of MTT signals was detected using an ELX800 microplate reader (BioTek Instruments, USA) at a wavelength of 570 nm. Fluorescence kits of Zip Alexa Fluor™ 555 (Thermo Fisher, USA) and Zip Alexa Fluor™ 488 (Thermo Fisher, USA) were used for the labeling of VEGF and BMP-2, respectively, as noted. To determine cell viability and proliferation activity, after 1 and 4 days of culture, the scaffolds containing hASCs were analyzed by SEM observation, LIVE/DEAD staining, and MTT cell viability analysis. In order to determine osteogenic activity, the ALP activity of the scaffold containing MC3T3-E1 cells was analyzed after 7 days of culture, and OCN expression and calcium deposition were analyzed after 21 days of culture. To determine neurogenesis activity, after 5 days of culture with electrical stimulation applied amplitude of 100 mV/cm for 1 h every day, samples containing PC12 cells were stained by immunofluorescence for the neuronal markers, Nestin and Tuj-1. The 2D fluorescence images were captured by using an ECLIPSE 80i fluorescence microscope (Nikon, Japan) equipped with Media Cybernetics Evolution VF Cooled Color Digital Cameras (VF-F-CLR-12-C), and performed by software (Image-Pro 6.2). The 3D fluorescence images were captured by using Leica confocal software, Leica Microsystems LAS AF (version 1.9.0), of the confocal laser scanning microscope (TCS SP5, Leica Microsystems, Germany) and reconstructed by using Imaris (version 9.7). Captured fluorescence images were analyzed by using ImageJ2 (beta version) for quantification of the protein marker expression and vascular organization, including the branch number and length[52]. Photoshop (version 19.0) was used to contrast and overlay images. The software, VK_Viewer (version VK-H1-V9), of VK-9500 3D profile microscope (Keyence, Japan) was used to collect laser confocal 3D/2D images. The 3D results were constructed by Keyence Analyzer (version VK-H1-A9).

**Cell co-culture.** Co-culture of cells: HUVECs and MG-63 cells were first stained with the noncytotoxic dyes CellTracker™ Green CMFDA (Thermo Fisher Scientific, USA) and PKH26 Red Fluorescent Cell Linker (Thermo Fisher Scientific, USA) following the manufacturer's instructions and subsequently encapsulated in different compartments of the patterned scaffold at an initial density of $1.5 \times 10^5$ and $0.5 \times 10^5$ cells/cm², respectively. In addition, hASCs were co-encapsulated with HUVECs in a ratio of 1:1 for the formation and maintenance of the tube network on the specified and fabricated scaffolds. The co-cultural samples were incubated with a culture medium that combined an equal volume of MEM-alpha (Thermo Fisher Scientific, USA) and M199 medium (Sigma-Aldrich, USA) and supplemented with 10% FBS (Biological Industries, Israel), 1% antibiotic–antimycotic (FBS, Biological Industries, Israel), $10 \times 10^{-3}$ M β-glycerol phosphate (Sigma-Aldrich, USA), 100 μg/mL L-ascorbic acid (Sigma-Aldrich, USA), $0.1 \times 10^{-6}$ M dexamethasone (Sigma-Aldrich, USA), 25 U/ml heparin (Sigma-Aldrich, USA), 30 μg/ml ECGS (Sigma-Aldrich, USA) for further characterization[48,49].

**Immunofluorescence staining.** Immunofluorescence staining was carried out according to the previously described protocol[53] with the following primary antibodies: anti-osteocalcin antibody (Abcam, Cat. No. ab198228, used at 1:50), Anti-Nestin antibody [Rat-401]–Neural Stem Cell Marker (Abcam, Cat. No.

ab6142, used at 1:500), Anti-beta III Tubulin antibody–Neuronal Marker (Abcam, Cat. No. ab18207, used at 1:500), Anti-CD31 antibody (Abcam, Cat. No. ab28364, used at 1:100), and Anti-Collagen I antibody [COL-1] (Abcam, Cat. No. ab6308, used at 1:200) and the following fluorescence-conjugated secondary antibodies: Goat Anti-Rabbit IgG H&L (Alexa Fluor® 488) preadsorbed (Abcam, Cat. No. ab150081, used at 1:200) and Goat Anti-Mouse IgG (whole molecule)−TRITC antibody (Sigma-Aldrich, Cat. No. T5393, used at 1:200). Cell nuclei were counterstained with 4',6-diamidino-2-phenylindole (Thermo Fisher Scientific, USA).

**Statistical analyses.** The data are reported as the mean value with standard deviation (mean ± SD), mentioned in each legend, from three independent samples, and unpaired $t$ test was used to determine the significance between data sets, with significance levels of $p < 0.05$ used as indicated.

**Reporting summary.** Further information on experimental design is available in the Nature Research Reporting Summary linked to this paper.

## Data availability
All relevant data generated or analyzed during this study are included in this published article and its supplementary information files or from the corresponding author upon reasonable request. A reporting summary for this Article is available as a Supplementary Information file. Source data are provided with this paper.

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

## Acknowledgements
H.-Y.C. gratefully acknowledges support from the Ministry of Science and Technology of Taiwan (MOST 108-2221-E-002-169-MY3 and 109-2314-B-002-041-MY3). This work was further supported by the "Advanced Research Center For Green Materials Science and Technology" from The Featured Area Research Center Program within the framework of the Higher Education Sprout Project by the Ministry of Education (109L9006) and the Ministry of Science and Technology in Taiwan (MOST 109-2634-F-002-042).

## Author contributions
H.-Y.C. designed the research; C.-Y.W., T.-Y.W., Z.-Y.G., Y.-C.Y., and T.-H.L. performed the research and analyzed the data; C.-Y.W., T.-Y.W., Z.-Y.G., P.-Y.W., Y.-C.Y., C.-W.H., and H.-Y.C. prepared the manuscript and contributed to the discussions of this study; H.-Y.C. acquired the financial support for the project leading to this publication.

## Competing interests
The authors declare no competing interests.
