## [Peer Review File · Nature Communications]

Reviewers' Comments:

Reviewer #1:

Remarks to the Author:

The authors have an interesting study about encapsulating biological materials into scaffolds using a vapor deposition technique. There is a lot of good work to show that the viability of the cells and biological uses such as co-cell culturing. The paper would be strengthened if the authors could add more details about the limitations of their fabrication process as discussed below:

- 1) About Figure 3, on line 210, the authors say "defined boundaries, which differentiated the compartments based on density and colour, were detected and corresponded to the shapes and dimensions." However, the boundaries do not seem very defined in the images in Figure 3 (there is a lot of diffusion). The authors should quantify the patterning limitations of the technique (for example, what is the difference between the ice template size and the resulting pattern size)? What limits the patterning?
- 2) What are the size limitations? How small can each compartment get and what limits this size?
- 3) How does the polymer deposition process and the sublimation process affect the scaffolding size?
- 4) On line 97, the authors say "based on the unique vapor sublimation and deposition mechanism." They cite their own work but they also need to cite earlier papers by Gupta and coworkers who demonstrated simultaneous vapor sublimation and deposition several years earlier (Macromolecules, 2013, 46, 2976, Journal of Vacuum Science & Technology A, 2014, 32, 041514.)

Reviewer #2:

Remarks to the Author:

The authors present a manuscript (NCOMMS-20-42730-T) entitled "Vapor-Phased Fabrication and Modulation of Cell-Laden Scaffolding Materials" and describe an interesting vapor-phase deposition to construct three-dimensional scaffold materials using modular iced templates and conditions for subsequent sublimation of ice and deposition of polychloro-p-xylylenes. Different combinations/shapes of modular components were evaluated by incorporation of multiple cell types (hASCs, MC3T3-E1, PC-12, HUVECs and MG-63) and biological signaling molecules (BMP-2, FGF-2, VEGF, PRP) with cell proliferation and tissue formation/fusion assessed following in vitro culture.

The Vapor-Phased Fabrication approach is interesting and provides a different angle on other similar modular approaches to generate combinations of osteogenic, vasculogenic and neuronal tissue modules in literature. While a new approach is described, there are many confusing aspects to the study around scaffold characterization and tissue differentiation, with the data presented not clearly backing up the claims that the authors suggest "collectively and unambiguously verified the hypotheses." Methodological steps described provide further confusion for readers wishing to replicate this work. The figures are lacking clarity and no evidence for statistical significance is provided. Specific comments addressing these aspects are provided below:

Major comments:

1. Figure 1 pg 10; text pg 4 –

- o There appears to be quite some red background staining in Fig1b. What is the % cell viability – are 50% cells alive or 90%? There is no quantitative analysis of cell viability which is vital for validation of the described new fabrication process given that it involves oil emulsions and liquid nitrogen freezing steps. Alamar blue and/or quantitative counts of live/dead cells are necessary to prove efficiency of the fabrication steps based against a well-established positive control fabrication process – e.g. cell encapsulated hydrogel (e.g. alginate, collagen) or pellet culture.
- o Fig 1e does not clearly show actin/phalloidin cytoskeleton filaments versus nucleus (no labels provided – assume PI cell nuclei are red?).
- o Why was this confirmed only on 1 cell type? What about other (more sensitive) cell types used with respect to cell viability and proliferation e.g. HUVECS. Was the iced template and sublimation process optimized for each cell type?

2. There is no confirmation if the oil from the oil-in-water cell suspension phase or the buffer

solvents were completely removed as part of the fabrication process. Did this effect cell viability and how could this impact clinical applicability of this approach? These aspects are not discussed.

3. No statistical significance is indicated for readers on Fig 2a (FGF v control), Fig 2b (effect of BMP-2 on OCN, ALP, Ca v control), Fig 2c PEDOT/PSS/PRP variants versus control for neurite length or neurogenic activity, Fig 4b (col I, nodule formation d7-21), Fig 4c (tube formation, branching d3-d10) or Fig S4b.

4. Fig 2 pg 11; text pg 4-5 –

- o Pg 5 – “Separate experiments and characterizations by using an MTT assay additionally showed statistically significant and consistent results, which suggested that the FGF-2-decorated samples resulted in an enhanced proliferation of hASCs.” There is no indication to readers what is statistically significant.

- o Fig 2a does not show any images of control samples. DNA content (or cell number) post seeding has not been confirmed to prove that cell content was equivalent between FGF and control samples to prove that observed differences were not from higher cell number in FGF conditions? Were the vapor phase fabricated structures (porosity, pore size) and mechanical properties of scaffolds identical between FGF and control conditions? i.e. did incorporation of FGF effect the sublimation process at all versus control? Without this confirmed, changes in scaffold properties may have accounted for changes in proliferation rate, not FGF. That extends to osteogenic and neuronal example tissues as well.

- o Fig 2a or caption does not indicate actin/phalloidin or PI staining for readers to interpret d1 versus d4 images.

- o Fig 2b, Fig 2c – as above, were cell content and scaffold properties unmodified by the addition of BMP2 modules for osteogenic differentiation, as well as PRP, PEDOT:PSS, PEDOT:PSS/PRP for neuronal differentiation?

5. Fig 3 pg 12; text pg 6-7 -

- o Pg 7 – the text discusses Compartment A versus Compartment B but there is no indication of which compartment is which in Fig 3a

- o Fig 3b - It's unclear what the 3D Profile images are trying to show. Clarity should be improved so readers can determine relationship with microCT and also 1a and 1c images.

- o There is no indication of staining in Fig 3c for HUVECS vs MG-63. What is green and what is red? What is cell nuclei and what is growth factor for each component? This is very confusing for readers and detracts from overall understanding of the specific models proposed. Fig S2 does not help provide further clarification as only fluoro VEGF and BMP-2 are shown with no color labels and appear in a different order to those images in Fig 3. These also appear to be different images to Fig 3 which is at odds with the text which describes these images as “merged” images. Fig S2b it is not clear what shapes/examples are trying to be achieved with labelled MG-63 cells. There is no labeling of HUVECS.

6. There is no information how mass production of scaffold modules was achieved for Fig S3. What are the 10mm cylindrical scaffolds in S3 illustrating?

7. Fig 4 pg 13; text pg 7-8 –

- o Fig 4 and claims also lack clarity. Fig 4 illustrates 2 compartments A and B for osteogenic and vasculogenic culture from separate MG-63/BMP2 (A) and HUVEC/hASC/VEGF (B) modules. There is no information on the size/shape of the components A and B. Was there a target geometry from one of the examples in Fig 3 that was attempted showing concurrent differentiation and tissue maturation within each of the components together? More critically there is no overview image proving both osteogenic and vascular components shown in Fig 4 have been cultured together as is inferred in text. An image should be provided showing the 2 co-cultured components and indicate the regions of that construct which form the panel of images for compartments A and B in Fig 4a, (similar to examples provided in Fig 3c).

- o No alpha-SMA (smooth muscle actin) staining is provided to show the role of ASCs in maturation of vessel networks in combination with HUVEC (CD31 stained), or other nuclear staining to show localization of ASCs. Images do not show mature networks or lumen claimed.

- o Fig 4c and text pg 8 - I do not believe the authors can claim to have measured “tube network formation” from the images provided or claim “A clear tube lumen started to form at day 7...” The images are not conclusive and Day 10 images look even less “vessel” like than at day 7. 10mm long vessels is extremely long and does not appear accurate – normally vessel length in normalized to surface area. Terminology for quantitative approach taken in Fig 4c should be changed to “network formation” and “network branch points”

8. Pg 7-8; Methods SI – There is insufficient description of the culture methodology. The only

information I could find describing culture conditions and media for all the different cell types for osteogenic, vascular and neural tissue formation was in SI pg 4: "The culturing conditions and media for cell culture were used according to the reported methods as referenced." It is not clear which references refer to this media and there is insufficient information for readers to repeat the methodological steps in this specific study.

o Furthermore in line with point 7 and point 8 above, co-culture of multiple tissue types requires complex combinations of individual media components (e.g. osteogenic and vasculogenic media) in order to support appropriate long-term (2-3 week) culture of merged tissue modules. There is no description of media to achieve examples Fig 4 for both combined osteogenic and vasculogenic differentiation over d10-d21. The fact that the compartments A and B were cultured for d21 and d10 each respectively seems to suggest that they were not co-cultured together as inferred. Further clarity and data is required to prove the claims in the study.

9. There is no discussion on how this proposed fabrication approach compares with other existing modular assembly approaches or describes drawbacks of the proposed strategy to put this work into context for readers. What is the benefit of this vapor phased approach to fabricate modular tissues offer compared to existing approaches for fabrication of complex 3D hierarchical tissues? The results of this study have relatively low impact given the previous work in 3D bioprinting of multi-material hydrogel bioinks. Previous examples including bottom-up assembly of preformed cell laden hydrogels (e.g. from Khademhosseini, Zhang) or 3D Bioprinting/biofabrication methods containing multimaterial bioinks and multicellular bioinks/modules/spheroids (e.g. Forgacs; Nakayama; Lutolf; Lewis).

Reviewer #3:

Remarks to the Author:

The authors fabricated 3D porous scaffolds with new functionalities by the addition of multiple dopants using a home-built sublimation/deposition system. They then tested their constructs for tissue engineering with multiple cells. In general, the discussion and results are well addressed, and the conclusions well supported by the results obtained. However, some issues, suggestions and curiosities are pointed out that might improve the quality of the manuscript:

1. The authors state "the iced template was formed by evaporating the water molecules, instead of depositing poly-p-xylylene polymer to replace the evaporated volume of water and forming a porous scaffold", which might lead to confusion. How is it the polymer controlled to be deposited at the same time is sublimated without been replaced? Is the speed of deposition an asset?
2. How long is the fabrication process from introduction of cells until incubation?
3. Please specify the color code in the immunofluorescence images.
4. Please add P-value and the number of repeats done for each experiment below the histograms to show the statistical significance differences and consistency of your results.
5. Please include controls of the culture of hASCs (Figure 2a), so the stated "enhanced attaching, spreading, and growth of hASCs on the porous structures" can be visualized.
6. Related to the "spreading and growth" of previous statement, according to Figure 2a, hASCs cells after 4 days of culture have the same roundish shape as in day 1, typical of unfold not extended cells, and no actin filaments can be distinguished. That can denote no functionality of the cells, i. e., no effective scaffolds for these cells. Please provide evidence of spreading, functionality and growth with other more specific function/proliferation markers, zoom-in images, etc.
7. What does "cell activity" in Figure 2a refer to? How is it calculated? Usually "activity" is synonym of "function", but according to the manuscript only an MTT assay has been done. May the authors have referred to viability instead? Please clarify.
8. Why ALP was evaluated after 7div while OCN analyses lasted 21 days? How many days of culture for the calcium analyses? Please include the images of ALP and calcium.
9. How many days were PC12 cultured in the scaffolds (Figure 2c)? Please include all the information related, including seeding density and media type and changes.
10. How does the doping of FGF-2, BMP-2, PRP, and PEDOT:PSS affects the scaffold's mechanical properties? It is strongly suggested to provide analyses of, at least, the Young's Modulus and swelling.
11. Why
12. Labels (i), (ii), (iii) have been used several times along the manuscript. What do these labels at lines 214-216 refer to? As well, which is compartment A and B in the scaffold's structure? A

schematic representation and inclusion of labels in Figure 3 will be appreciated to facilitate understanding.

13. Which is the final aim of the co-cultures in separate compartments? And why have the authors chosen those specific cell lines?

We would like to thank all reviewers of our manuscript for their comments and suggestions for improvement on our manuscript. In the following, we will address all comments and explain our rational and resulting changes to the manuscript in detail. The original statements of the reviewers are shown in plain black, our responses in bold red. For easiness of follow-up, we marked the changes yellow in the main manuscript.

Response to Reviewer 1:

Comments:

The authors have an interesting study about encapsulating biological materials into scaffolds using a vapor deposition technique. There is a lot of good work to show that the viability of the cells and biological uses such as co-cell culturing. The paper would be strengthened if the authors could add more details about the limitations of their fabrication process as discussed below:

We appreciate the comments of the reviewer and are very delighted with the positive feedback.

1) About Figure 3, on line 210, the authors say "defined boundaries, which differentiated the compartments based on density and colour, were detected and corresponded to the shapes and dimensions." However, the boundaries do not seem very defined in the images in Figure 3 (there is a lot of diffusion). The authors should quantify the patterning limitations of the technique (for example, what is the difference between the ice template size and the resulting pattern size)? What limits the patterning?

Ans:

We appreciate the comments of the reviewer. We have performed additional statistical analysis with the ice templates and the fabricated modules and scaffolds in the studies, as suggested by the reviewer. The analysis was performed based on comparing the ice template modules (**Figure S5-a**), scaffold products (**Figure S5-b**), and the compartmentalized modules within one modulated scaffolds (**Figure 3**, 3-D profile and micro-CT images). Briefly, the accuracy could be precisely controlled above approximately 95% for micro-sized module products and above 99% for large scale (mm) modules; these additional data are now included in the revised Supplementary Materials in **Table S1**. We have also included an additional discussion in **page 7**, with respect to the precision limitation of the proposed fabrication method. Furthermore, high-quality images by micro-CT and microscope are also included in the revised **Figure 3** showing better contrast at the boundaries. The additional and revised data are also shown below for the review:

In page 7:

..... A >95% accuracy of the compartmentalized modules within one modulated scaffold construct was estimated, and the detailed dimensional data comparing the ice templates and the scaffold modules were summarized in the Supplementary Materials in Table S1.

Table S1:

Table S1. Dimensional specifications and comparisons of the ice templates and the scaffold modules.

	Ice Template		Scaffold Modules				Modulated Scaffold			
	edge/ diameter	height	edge/ diameter	height	accuracy (edge/ diameter)	accuracy (height)	edge/ diameter	height	accuracy (edge/ diameter)	accuracy (height)
Discontinued configuration (μm)	283 \pm 5	280 \pm 7	279 \pm 7	271 \pm 6	98.6%	96.8%	281 \pm 6	281 \pm 10	99.3%	99.6%
Asymmetrical (μm)	311 \pm 5	247 \pm 10	299 \pm 9	239 \pm 9	96.1%	96.8%	309 \pm 4	240 \pm 8	99.4%	97.2%
Cylindrical structure (mm)	5.00 \pm 0.06	4.91 \pm 0.01	4.99 \pm 0.07	4.88 \pm 0.10	99.8%	99.4%	-	-	-	-

Figure 3. Assembly of various modules with defined chemical/biological components and geometric architecture into one modulated construct with different compartment A vs. B. Components used for assembly and fabrication of the two compartments included (a) Oil Red-O (red channel) vs. fluorescein-5-isothiocyanate (green channel), (b) blank solution vs. silver (Ag) nanoparticles in 3-D profiled images and tomographic images, and (c) co-cultured system HUVECs combined with VEGF vs. MG-63 and BMP-2. The geometric illustration of the assembly and fabrication is shown on the left indicating equal aspect ratio, asymmetrical, discontinued configuration, and curved distribution of the two compartments.

(a) ice templates

vapor fabrication process

(b) scaffold modules

Figure S5. Mass production to fabricate the (a) ice templates and (b) scaffold modules with various shapes and sizes. The shape, size, and number of features was determined on a PDMS mold to produce the ice templates and the vapor fabrication process was then performed to result in the final scaffold modules. The ice templates were produced from solutions containing blue dyes for the purpose of visualization. The images of (a) and (b) were captured from different batch of samples.

2) What are the size limitations? How small can each compartment get and what limits this size?

Ans:

We appreciate the comments of the reviewer. The reviewer raises an excellent point regarding the size limitations. Theoretically, the size limitation is set on the ice templates (the size and shape of fabricated scaffold products replicate ice templates with high precision, as discussed in previous Q1), and with current technologies to size and shape the ice templates using molding techniques, micrometer to submicron sized molds were reported [Uriarte et al., *Journal of Mechanical Engineering Science* 2006, 220, 1665; Cottet et al., *Biomeicrofluidics* 2017, 11, 064111]. In addition, a breakthrough of the size limitation by the reported fabrication method can

be performed based on a stage-wise control and timing of the involved sublimation and deposition processes, were also previously reported by our group [Tung et al., *Nature Communications* 2018, 9, 2564; Wu et al., *Coatings* 2020, 10, 1248], and a particle size below 50 nanometers was shown. We have also, in the revised manuscript in **page 4**, included an additional discussion regarding the size limitation issue, as suggested by the reviewer. The additional discussion is also shown below for the review:

In page 4:

..... The fabrication is a time-dependant process to produce a proportional volume of scaffold product and required approximately 60 mins for a 5 cm³-sized sample, and theoretically, size of the fabricated scaffold module is limited to the ice templates that can be produced by existing techniques, and can be tuned based on a stage-wise control and timing of the involved sublimation and deposition processes.

3) How does the polymer deposition process and the sublimation process affect the scaffolding size?

Ans:

In response to the reviewer's comments, based on the previous discussions in Q1 and Q2, the overall size and shape of the fabricated scaffold products are not affected by the polymer deposition process and the sublimation process, however the internal structures of the scaffolds including pore size and porosity are affected. The caused mechanism and the controllability of the internal structure were explained and attempted in our previous reports mainly due to the dopants in the fabrication process. The dopants were tried including inorganic Au, Ag, Fe oxides, carbon dioxide, and also organic solvents, such as methanol, ethanol, acetone, hexane. Briefly, these dopants were classified (by us) into two categories of "volatile" and "non-volatile" during the processing thermodynamic conditions (approximately 0.1 bar and room temperature or below), and we found the fabricated porous polymer products were greatly impacted with their mechanical properties, e.g. pore size, porosity, Young's modulus, and etc., by "volatile" dopants and was found due to the sublimation (volatility) during the fabrication process [Tung et al., *Applied Materials Today* 2017, 7, 77; Chiu et al., *Chemistry of Materials* 2020, 32, 1120], and these mechanical properties were tunable to show in a wide range, for instance, pore sizes ranging from >5 μm to 100 μm, and porosities of approximately 50% - 80%, Young's modulus from ~100 kPa to 10000 kPa, were attempted and tuned. By contrast, the "non-volatile" dopants were showing negligible impact to the mechanical properties of fabricated porous products, due to (i) non-sublimating behavior, and (ii) small amount of dopants compare to the polymer matrix;

and the overall mechanical properties were assume consistent to a blank porous product (only with polymer matrix). Considering the dopants used in the current study, e.g. FGF-2, BMP-2, PRP, and PEDOT:PSS were in the “non-volatile” category and were used in relatively small amount compared to the polymer scaffold matrix, consistent and negligible impact of these dopants to the overall mechanical properties are expected. Additional measurements of these properties were performed and were discussed in the revised manuscript in **page 2-3** and **page 4** as suggested by the reviewer. The additional discussions are also shown below for the review:

In page 2-3:

.....With using volatile compounds as dopants, the mechanical properties of the resultant porous materials were tuneable in a wide range, for instance, pore sizes ranging from >5 μm to 100 μm , and porosities of approximately 50% - 80%, Young’s modulus from ~10 kPa to 10000 kPa, were attempted and tuned; while the use of non-volatile dopants render a uniform or controlled localization of encapsulating the dopants and without interference to the mechanical properties.

And

In page 4:

..... With also tuneable mechanical properties to fabricate the scaffold modules,²¹ consistent properties including approximately $35.7 \pm 8.2 \mu\text{m}$ in pore size, $63.4\% \pm 6.3$ porosity, and $150 \pm 21.5 \text{ kPa}$ for the Young’s modulus, were measured and used in the current studies.

4) On line 97, the authors say "based on the unique vapor sublimation and deposition mechanism." They cite their own work but they also need to cite earlier papers by Gupta and coworkers who demonstrated simultaneous vapor sublimation and deposition several years earlier (Macromolecules, 2013, 46, 2976, Journal of Vacuum Science & Technology A, 2014, 32, 041514.)

Ans:

We appreciate the comments of the reviewer. The suggested references are cited in the revised manuscript in **page 4** [ref. 23 – 24].

Response to Reviewer 2:

Comments:

The authors present a manuscript (NCOMMS-20-42730-T) entitled “Vapor-Phased Fabrication and Modulation of Cell-Laden Scaffolding Materials” and describe an interesting vapor-phase deposition to construct three-dimensional scaffold materials using modular iced templates and conditions for subsequent sublimation of ice and deposition of polychloro-p-xylylenes. Different combinations/shapes of modular components were evaluated by incorporation of multiple cell types (hASCs, MC3T3-E1, PC-12, HUVECs and MG-63) and biological signaling molecules (BMP-2, FGF-2, VEGF, PRP) with cell proliferation and tissue formation/fusion assessed following in vitro culture.

The Vapor-Phased Fabrication approach is interesting and provides a different angle on other similar modular approaches to generate combinations of osteogenic, vasculogenic and neuronal tissue modules in literature. While a new approach is described, there are many confusing aspects to the study around scaffold characterization and tissue differentiation, with the data presented not clearly backing up the claims that the authors suggest “collectively and unambiguously verified the hypotheses.” Methodological steps described provide further confusion for readers wishing to replicate this work. The figures are lacking clarity and no evidence for statistical significance is provided. Specific comments addressing these aspects are provided below:

We appreciate the comments of the reviewer and are very delighted with the positive feedback.

1. Figure 1 pg 10; text pg 4 –

o There appears to be quite some red background staining in Fig1b. What is the % cell viability – are 50% cells alive or 90%?

Ans:

We appreciate the comments of the reviewer. The red background in **Figure 1d** was the reflection of noise signals from the scaffold surface; the pattern of these noise signals can be compared to the background seen in **Figure 1e**, which very similar background pattern of the scaffold can be identified. The dead cells were stained with much brighter red signals. We have also, in the revised manuscript, included additional calculated results of the cell viability data, as suggested by the reviewer. Plus, we have included additional cell viability experiments in different stages during the fabrication process, and these new data are included in the Supplementary Materials in **Figure S1**. Briefly, >80% of viable cells were estimated for the laden cells during the fabrication process in the study. An additional discussion regarding the cell

viability due to the fabrication process was added in **page 4** in order for the better understanding of readers. The additional data and discussions are also shown below for the review:

Figure S1. Quantitative analysis of the cell viability during scaffold fabrication process. (a) Schematic illustration of fabrication stages. LIVE/DEAD staining technique was performed by showing live cells in green channel and dead cells in red channel, and the signals were calculated to determine the cell viability. The recorded fluorescence images and the calculated survival rates in different stages were shown in (b) oil-in-water suspension, 98.2%; (c) after freezing in ice templates, 96.1%; and (d) after fabrication in the final scaffold constructs, 80.8%.

In page 4:

..... An overall >80% rate of viable cells was estimated based on comparing the LIVE/DEAD signals, including 98.2% of viable cells in oil-in-water suspension, 96.1% after freezing in the ice templates, and 80.8% after fabrication in the final scaffold constructs. Additional data of analysing the cell viability are also included in the Supplementary Materials in Figure S1.

There is no quantitative analysis of cell viability which is vital for validation of the described new fabrication proves given that it involves oil emulsions and liquid nitrogen freezing steps. Alamar blue and/or quantitative counts of live/dead cells are necessary to prove efficiency of the fabrication steps based against a well-established positive control fabrication process – e.g. cell encapsulated hydrogel (e.g. alginate, collagen) or pellet culture.

Ans:

We appreciate the comments of the reviewer. We have, performed additional quantitative analysis of cell viability for the proposed fabrication process, and the new data are included in **Figure S1** and were discussion in **page 4**, as suggested by the reviewer. Briefly, the cells were protected in the oil-in-water suspension system during the freezing and fabrication process, and the survival rate of cells is quantified by counting of live/dead cells, and showed approximately 98.2% (in oil-in-water suspension), 96.1% (after freezing in ice templates), and 80.8% (after fabrication in the final scaffold constructs). The additional data are also shown below for the review:

Figure S1. Quantitative analysis of the cell viability during scaffold fabrication process. (a) Schematic illustration of fabrication stages. LIVE/DEAD staining technique was performed by showing live cells in green channel and dead cells in red channel, and the signals were calculated to determine the cell viability. The recorded fluorescence images and the calculated survival rates in different stages were shown in (b) oil-in-water suspension, 98.2%; (c) after freezing in ice templates, 96.1%; and (d) after fabrication in the final scaffold constructs, 80.8%.

In page 4:

..... An overall >80% rate of viable cells was estimated based on comparing the LIVE/DEAD signals, including 98.2% of viable cells in oil-in-water suspension, 96.1% after freezing in the ice templates, and 80.8% after fabrication in the final scaffold constructs. Additional data of analysing the cell viability are also included in the Supplementary Materials in Figure S1.

o Fig 1e does not clearly show actin/phalloidin cytoskeleton filaments versus nucleus (no labels provided – assume PI cell nuclei are red?).

Ans:

We appreciate the comments of the reviewer. We have, in the revised manuscript, included a higher resolution image of **Figure 1e** and indications of the staining dyes to better described the cytoskeleton filaments and nucleus in the captions, as suggested by the reviewer. The revised **Figure 1** is also shown below for the review:

Figure 1. Fabrication of cell-laden and multifunctional scaffold modules. (a) Schematic illustration of vapor-phase sublimation and deposition fabrication process. Iced template modules with a defined internal composition and outer architecture were formed through discrete module assembly and resulted in a continuous final modulated construct by subsequent vapor deposition. (b) The analysis of vapor composition by real-time mass spectrometry during the fabrication process shows the presence of sublimated water vapor at 18 amu and the deposited quinodimethane monomers at 104 amu and 139 amu. (c) An overlaid image of LIVE/DEAD-stained hASCs prepared in an oil-in-water system. Verification of hASCs within a fabricated module after vapor-phase deposition process by (d) LIVE/DEAD staining, (e) fluorescence labelling with Alexa Fluor® 488-conjugated phalloidin to stain the cytoskeleton (green) and propidium iodide to stain the nucleus (red); a high resolution and magnified image of a stained cell was shown in the inset, and (f) SEM dissection image.

o Why was this confirmed only on 1 cell type? What about other (more sensitive) cell types used with respect to cell viability and proliferation e.g. HUVECS. Was the iced template and sublimation process optimized for each cell type?

Ans:

In response to the reviewer's comments, the material, poly-p-xylylene, used to fabricate the scaffold is a USP (United States Pharmacopeia) Class VI highly biocompatible polymer, and the cell viability/biocompatibility examinations associated with the polymer system (thin films) or the same produced devices were well-reported during the past including biocompatibility against various types of cells (including HUVEC) [Tsai et al., *Advanced Functional Materials* 2014, 24, 2281; Wu et al., *Colloids and Surfaces B: Biointerfaces* 2019, 175, 545; Wu et al., *Chemistry of Materials* 2015, 27, 7028], types of stem cells [Wu et al. *Advanced Materials Interfaces* 2017, 1700243; Chen et al., *ACS Applied Materials & Interfaces* 2018, 10, 31882], and immunological compatibilities [Wu et al., *Materials Science and Engineering: C* 2016, 69, 283; Hsu et al., *Scientific Reports* 2019, 9, 7644]. We therefore suggest keep the structure of using hASC as the model system to demonstrate the cell viability and use citations to address the applicability and extended compatibility to other types of cells that are related to the scaffold materials. Plus, the following demonstrations used in the current report have also ambiguously supported these cell viability issues. We have included additional discussions to address the cellular compatibility with respect to the polymer materials and fabrication process, in the revised manuscript in **page 5**. The additional discussions are also shown below for the review:

In page 5:

.....The cell viabilities and compatibilities are expected extendable to various types of cell and stem cell systems by using the fabrication process and the poly-p-xylylene (and the derivatives) scaffold materials based on the compatibilities studies reported elsewhere [ref. 25-28].

2. There is no confirmation if the oil from the oil-in-water cell suspension phase or the buffer solvents were completely removed as part of the fabrication process. Did this effect cell viability and how could this impact clinical applicability of this approach? These aspects are not discussed.

Ans:

We appreciate the comments of the reviewer. In response to the reviewer's comments, the oil-in-water suspension system was used in our previous study with loading fluorescence molecules (Alexa Fluor 633, red signals) as the dopants for the purpose of fabrication and decoration of the interior porous structure for a polymer particle [Chiu et al. *Chemistry of Materials* 2020, 32, 1120, in Figure 3 and the discussions], and the results showed a branched distribution of footprint left by the red signals which was discovered attributed to the pathway of diffusion by the non-sublimating oil that was removed from the porous particle material. In addition, the glyceryl trioleate oil used in the current was chosen purposely due to its reported cellular biocompatibility [Mosca et al., *The Journal of Physical Chemistry B* 2006, 110, 25994-26000; Desai et al., *J Pharm Sci* 2020, 109, 1752-1764], and compatible cells with excellent cell viability were also found in the current study (in **Figure S1**, and data in **page 4**). We thus, in the revised manuscript in **page 4**, included additional discussions regarding the impact of oil-in-water setup to the scaffold and the cell viability; relevant references were also cited. The additional data and discussions are also shown below for the review:

In page 4:

.....Removal of the oil from the scaffold system was expected based on previously discovered results [ref. 21]. An overall >80% rate of viable cells was estimated based on comparing the LIVE/DEAD signals, including 98.2% of viable cells in oil-in-water suspension, 96.1% after freezing in the ice templates, and 80.8% after fabrication in the final scaffold constructs. The cell viabilities and compatibilities are expected extendable to various types of cell and stem cell systems by using the fabrication process and the poly-p-xylylene (and the derivatives) scaffold materials based on the compatibilities studies reported elsewhere.

Figure S1. Quantitative analysis of the cell viability during scaffold fabrication process. (a) Schematic illustration of fabrication stages. LIVE/DEAD staining technique was performed by showing live cells in green channel and dead cells in red channel, and the signals were calculated to determine the cell viability. The recorded fluorescence images and the calculated survival rates in different stages were shown in (b) oil-in-water suspension, 98.2%; (c) after freezing in ice templates, 96.1%; and (d) after fabrication in the final scaffold constructs, 80.8%.

3. No statistical significance is indicated for readers on Fig 2a (FGF v control), Fig 2b (effect of BMP-2 on OCN, ALP, Ca v control), Fig 2c PEDOT/PSS/PRP variants versus control for neurite length or neurogenic activity, Fig 4b (col I, nodule formation d7-21), Fig 4c (tube formation, branching d3-d10) or Fig S4b.

Ans:

We appreciate the comments of the reviewer. We have, in the revised manuscript, included statistical significance indications for all the relevant data including **Figure 2a** (FGF v control), **Figure 2b** (effect of BMP-2 on OCN, ALP, Ca v control), **Figure 2c** PEDOT/PSS/PRP variants versus control for neurite length or neurogenic activity, **Figure 4b** (col I, nodule formation d7-

21), **Figure 4c** (tube formation, branching d3-d10), and **Figure S6** (due to the addition of new data, the numbering of Figure S4b is now become Figure S6b), as suggested by the reviewer, and also for the better understanding of readers. The revised **Figures** of data are also shown below for the review:

Figure 2. Fabrication of various modules with specified cell-guiding activities. (a) SEM and laser confocal images were recorded for modules containing hASCs cells and FGF-2 and showed cell proliferation activities after culturing for 4 days. The F-actin cytoskeleton (green) was stained by Alexa Fluor® 488-conjugated phalloidin and the nucleus (red) was stained with propidium iodide; the inset images showed attached and spreaded cells. (b) Modules composed of MC3T3-E1 cells and BMP-2 showed guided osteogenesis activities. The ALP expression was analyzed after

7 days of culture, while OCN expression and calcium deposition were analyzed after 21 days of culture. (c) Modules composed of more complex components, including PC12 cells, PRP molecules, and the PEDOT:PSS system, were fabricated and showed guided neurogenesis activities after 5 days of culture. Data are expressed as the mean value with the standard deviation based on three independent samples. Significance levels are indicated according to the unpaired t-test (n.s., nonsignificant difference; * $p < 0.05$; ** $p < 0.01$; and *** $p < 0.001$).

Figure 4. Cell co-cultures of the modulated constructs with defined and customized compartment of A containing the combination of MG-63 and BMP-2, and compartment B containing HUVECs, hASCs, and VEGF. The fabricated scaffold samples with the configuration of an equal aspect ratio for compartment A vs. compartment B was chosen in the study. (a) The recorded immunofluorescence images of fibrous type-I collagen (COL-I) expression at day 7, day 14, and day 21 indicated guided osteogenic activities in compartment-A during the co-cultures. The bone-like nodules formed and increased in size over the indicated period. On the other hand, representative immunofluorescence images of platelet endothelial cell adhesion molecule (PECAM-1/CD31) expression at day 3, day 7, and day 10 showed guided angiogenic activities in compartment B. The classic tube network formed and developed increasingly over the indicated period. (b) Quantitative analysis of osteogenic activities and (c) angiogenic activities of the MG-63 and HUVEC co-culture samples containing compartmentalized BMP-2 and VEGF in compartment A and compartment B, respectively. Statistical analysis was based on three independent experiments with three duplicated samples, and each bar represents the mean value and the standard deviation (\pm SD). Significance levels are indicated according to the unpaired t-test analysis (* $p < 0.05$; and ** $p < 0.01$).

Figure S6. (a) Immunofluorescence images show the specific expression of the protein markers on MG-63 (human osteoblasts) and HUVECs (human umbilical vein endothelial cells). COL-I, type-I collagen; OCN, osteocalcin; CD31, cluster of differentiation 31 (platelet endothelial cell adhesion molecule); and VEGFR2, vascular endothelial growth factor receptor-2. (b) Quantitative analysis of the protein marker expression on MG-63 and HUVECs not only confirmed the cell identities but also verified the methodology to visualize MG-63 and HUVECs specifically in the fabricated modulated scaffolds. Statistical results were analysed based on three independent experiments of three duplicated samples, and each bar represents the mean value and the standard deviation (\pm SD). The significance level is shown by the unpaired t-test. *** $p < 0.001$.

4. Fig 2 pg 11; text pg 4-5 –

o Pg 5 – “Separate experiments and characterizations by using an MTT assay additionally showed statistically significant and consistent results, which suggested that the FGF-2-decorated samples resulted in an enhanced proliferation of hASCs.” There is no indication to readers what is statistically significant.

Ans:

We appreciate the comments of the reviewer. We have, in the revised manuscript, included statistical significance indications, as suggested by the reviewer. The revised **Figure 2** is also shown below for the review:

Figure 2. Fabrication of various modules with specified cell-guiding activities. (a) SEM and laser confocal images were recorded for modules containing hASCs cells and FGF-2 and showed cell proliferation activities after culturing for 4 days. The F-actin cytoskeleton (green) was stained by Alexa Fluor® 488-conjugated phalloidin and the nucleus (red) was stained with propidium iodide;

the inset images showed attached and spreaded cells. (b) Modules composed of MC3T3-E1 cells and BMP-2 showed guided osteogenesis activities. The ALP expression was analyzed after 7 days of culture, while OCN expression and calcium deposition were analyzed after 21 days of culture. (c) Modules composed of more complex components, including PC12 cells, PRP molecules, and the PEDOT:PSS system, were fabricated and showed guided neurogenesis activities after 5 days of culture. Data are expressed as the mean value with the standard deviation based on three independent samples. Significance levels are indicated according to the unpaired t-test (n.s., nonsignificant difference; * $p < 0.05$; ** $p < 0.01$; and *** $p < 0.001$).

o Fig 2a does not show any images of control samples. DNA content (or cell number) post seeding has not been confirmed to prove that cell content was equivalent between FGF and control samples to prove that observed differences were not from higher cell number in FGF conditions?

Ans:

We appreciate the comments of the reviewer. We have performed additional control experiments as suggested by the reviewer. These additional data include images from SEM and confocal microscope results to compare the FGF-2-decorated samples with samples without FGF-2, and are shown in the revised **Figure 2**, and the cell numbers are also statistically compared. Furthermore, additional Live/Dead staining images are included in the Supplementary Materials in **Figure S2** to indicate the cell population were equivalent between the FGF-2-decorated samples and control samples at day 1, while higher cell number was found for the FGF-2-decorated group at day 4. The revised **Figure 2**, additional discussions, and **Figure S2** are also shown below for the review:

Figure 2. Fabrication of various modules with specified cell-guiding activities. (a) SEM and laser confocal images were recorded for modules containing hASCs cells and FGF-2 and showed cell proliferation activities after culturing for 4 days. The F-actin cytoskeleton (green) was stained by Alexa Fluor® 488-conjugated phalloidin and the nucleus (red) was stained with propidium iodide; the inset images showed attached and spreaded cells. (b) Modules composed of MC3T3-E1 cells and BMP-2 showed guided osteogenesis activities. The ALP expression was analyzed after 7 days of culture, while OCN expression and calcium deposition were analyzed after 21 days of culture. (c) Modules composed of more complex components, including PC12 cells, PRP molecules, and the PEDOT:PSS system, were fabricated and showed guided neurogenesis activities after 5 days of culture. Data are expressed as the mean value with the standard deviation based on three independent samples. Significance levels are indicated according to the unpaired t-test (n.s., nonsignificant difference; * $p < 0.05$; ** $p < 0.01$; and *** $p < 0.001$).

In page 5:

..... Additional control experiments were performed by Live/Dead staining showing the comparison of the FGF-2-decorated samples with samples without FGF-2 at day 1 and day 4, and the data are included in the Supplementary Materials in Figure S2.

Figure S2. (a) Fluorescence images by Live/Dead staining comparing the FGF-2-decorated samples and the control group of samples (without FGF-2). (b) Comparison of the calculated live cell number revealing the cell population were equivalent between the FGF-2-decorated samples and control samples at day 1, while higher cell number (** $p < 0.001$) was found for the FGF-2-decorated group at day 4. (c) Cell viability analysis based on the ratio of total viable cells/total cells revealed both FGF-2-decorated and control samples have a good biocompatibility with no statistically significant difference. Data were analysed based on three independent experiments of three duplicated samples, and each bar represents the mean value and the standard deviation (\pm SD). Significance levels are shown by the unpaired t-test. n.s., nonsignificant difference.

Were the vapor phase fabricated structures (porosity, pore size) and mechanical properties of scaffolds identical between FGF and control conditions? i.e. did incorporation of FGF effect the sublimation process at all versus control? Without this confirmed, changes in scaffold properties may have accounted for changes in proliferation rate, not FGF. That extends to osteogenic and neuronal example tissues as well.

Ans:

We appreciate the comments of the reviewer. In our previous studies, dopants were tried including inorganic Au, Ag, Fe oxides, carbon dioxide, and also organic solvents, such as methanol, ethanol, acetone, hexane. Briefly, these dopants were classified (by us) into two categories of “volatile” and “non-volatile” during the processing thermodynamic conditions (approximately 0.1 bar and room temperature or below), and we found the fabricated porous polymer products were greatly impacted with their mechanical properties, e.g. pore size, porosity, Young’s modulus, and etc., by “volatile” dopants and was found due to the sublimation (volatility) during the fabrication process [Tung et al. *Applied Materials Today* 2017, 7, 77; Chiu et al. *Chemistry of Materials* 2020, 32, 1120], and these mechanical properties were tunable to show in a wide range, for instance, pore sizes ranging from >5 μm to 100 μm , and porosities of approximately 50% - 80%, Young’s modulus from ~100 kPa to 10000 kPa, were attempted and tuned. By contrast, the “non-volatile” dopants were showing negligible impact to the mechanical properties of fabricated porous products, due to (i) non-sublimating behavior, and (ii) small amount of dopants compare to the polymer matrix; and the overall mechanical properties were assume consistent to a blank porous product (only with polymer matrix). Considering the dopants used in the current study, e.g. FGF-2, BMP-2, PRP, and PEDOT:PSS were in the “non-volatile” category and were used in relatively small amount compared to the polymer scaffold matrix, consistent and negligible impact of these dopants to the overall mechanical properties are expected. Additional measurements of these properties were performed and were discussed in the revised manuscript in **page 2-3** and **page 4**, as suggested by the reviewer. The additional discussions are also shown below for the review:

In page 2-3:

..... With using volatile compounds as dopants, the mechanical properties of the resultant porous materials were tuneable in a wide range, for instance, pore sizes ranging from >5 μm to 100 μm , and porosities of approximately 50% - 80%, Young’s modulus from ~100 kPa to 10000 kPa, were attempted and tuned; while the use of non-volatile dopants render a uniform or controlled localization of encapsulating the dopants and without interference to the mechanical properties.

And

In page 4:

..... With also tuneable mechanical properties to fabricate the scaffold modules,²¹ consistent properties including approximately $35.7 \pm 8.2 \mu\text{m}$ in pore size, $63.4\% \pm 6.3$ porosity, and $150 \pm 21.5 \text{ kPa}$ for the Young's modulus, were measured and used in the current studies.

o Fig 2a or caption does not indicate actin/phalloidin or PI staining for readers to interpret d1 versus d4 images.

Ans:

We appreciate the comments of the reviewer. The suggested indications of fluorescent phalloidin and PI labels representing the actin filaments (green) and the nucleus (red) are now included in the caption in the revised **Figure 2**, and are also shown below for the review:

Figure 2. Fabrication of various modules with specified cell-guiding activities. (a) SEM and laser confocal images were recorded for modules containing hASCs cells and FGF-2 and showed cell proliferation activities after culturing for 4 days. The F-actin cytoskeleton (green) was stained by Alexa Fluor® 488-conjugated phalloidin and the nucleus (red) was stained with propidium iodide; the inset images showed attached and spreaded cells. (b) Modules composed of MC3T3-E1 cells and BMP-2 showed guided osteogenesis activities. The ALP expression was analyzed after 7 days of culture, while OCN expression and calcium deposition were analyzed after 21 days of culture. (c) Modules composed of more complex components, including PC12 cells, PRP molecules, and the PEDOT:PSS system, were fabricated and showed guided neurogenesis activities after 5 days of culture. Data are expressed as the mean value with the standard deviation based on three independent samples. Significance levels are indicated according to the unpaired t-test (n.s., nonsignificant difference; * $p < 0.05$; ** $p < 0.01$; and *** $p < 0.001$).

o Fig 2b, Fig 2c – as above, were cell content and scaffold properties unmodified by the addition of BMP2 modules for osteogenic differentiation, as well as PRP, PEDOT:PSS, PEDOT:PSS/PRP for neuronal differentiation?

Ans:

In response to the reviewer's comments, and similar to the answers described previously; the dopants used in the current study, e.g. FGF-2, BMP-2, PRP, and PEDOT:PSS were in the "non-volatile" category, and the "non-volatile" dopants were showing negligible impact to the mechanical properties of fabricated porous products, due to (i) non-sublimating behavior, and (ii) small amount of dopants compare to the polymer matrix; and the overall mechanical properties were assume consistent to a blank porous product (only with polymer matrix). The fabrications of scaffold materials with using varied non-volatile dopants to render consistent mechanical properties are expected. Additional measurements of these properties were performed and were discussed in the revised manuscript in **page 2-3** and **page 4**, as suggested by the reviewer. The additional discussions are also shown below for the review:

In page 2-3:

..... With using volatile compounds as dopants, the mechanical properties of the resultant porous materials were tuneable in a wide range, for instance, pore sizes ranging from >5 μm to 100 μm , and porosities of approximately 50% - 80%, Young's modulus from ~100 kPa to 10000 kPa, were attempted and tuned; while the use of non-volatile dopants render a uniform or controlled localization of encapsulating the dopants and without interference to the mechanical properties.

And

In page 4:

..... With also tuneable mechanical properties to fabricate the scaffold modules, 21 consistent properties including approximately $35.7 \pm 8.2 \mu\text{m}$ in pore size, $63.4\% \pm 6.3$ porosity, and $150 \pm 21.5 \text{ kPa}$ for the Young's modulus, were measured and used in the current studies.

5. Fig 3 pg 12; text pg 6-7 -

o Pg 7 – the text discusses Compartment A versus Compartment B but there is no indication of which compartment is which in Fig 3a.

Ans:

We appreciate the comments of the reviewer. we have, in the revised manuscript, included additional indications to better define the compartment A and B in **page 7-8** and in **Figure 3**, as suggested by the reviewer. The addition changes and the revised **Figure 3** are also shown below for the review:

In page 7:

.....and the same vapor construction process transformed the assembled iced templates to produce separate compartments of A and B with varied geometries and dimensions in one modulated scaffold. Demonstrations were shown with the versatility to configure compartment A vs. B in equal aspect ratio, asymmetrical, discontinued configuration, and curved distribution of these two compartments.

And

In page 8:

..... Calculated volume ratios for the studied configurations based on analysing the 3D-profiled and micro-CT results were also obtained showing approximately 50% vs. 50% (equal aspect ratio), 56% vs. 44% (asymmetrical), 98% vs. 2% (discontinued configuration), and 26% vs. 74% (curved distribution) for compartment A vs. compartment B, respectively.

Figure 3. Assembly of various modules with defined chemical/biological components and geometric architecture into one modulated construct with different compartment A vs. B. Components used for assembly and fabrication of the two compartments included (a) Oil Red-O (red channel) vs. fluorescein-5-isothiocyanate (green channel), (b) blank solution vs. silver (Ag) nanoparticles in 3-D profiled images and tomographic images, and (c) co-cultured system HUVECs combined with VEGF vs. MG-63 and BMP-2. The geometric illustration of the assembly and fabrication is shown on the left indicating equal aspect ratio, asymmetrical, discontinued configuration, and curved distribution of the two compartments.

o Fig 3b - It's unclear what the 3D Profile images are trying to show. Clarity should be improved so readers can determine relationship with microCT and also 1a and 1c images.

Ans:

In response to the reviewer's comments, the 3D profile images intended to render an overview of the configuration of compartments of A and B within one scaffold construct, and in addition to

demonstrate the devised configurations including equal aspect ratio, asymmetrical, discontinued configuration, and curved distribution of these two compartments, information regarding the volume ratio was calculated based on 3D profile results and the micro-CT results, and approximately 50% vs. 50% (equal aspect ratio), 56% vs. 44% (asymmetrical), 98% vs. 2% (discontinued configuration), and 26% vs. 74% (curved distribution) for compartment A vs. compartment B, respectively, were estimated. We have, in the revised **Figure 3**, included images with higher resolution of these 3D profile images, as suggested by the reviewer, and also included additional discussions regarding the configuration and volume ration information obtained by the 3D profile results. The revised **Figure 3** and additional discussions (**page7** and **8**) are also shown below for the review:

Figure 3. Assembly of various modules with defined chemical/biological components and geometric architecture into one modulated construct with different compartment A vs. B. Components used for assembly and fabrication of the two compartments included (a) Oil Red-O (red channel) vs. fluorescein-5-isothiocyanate (green channel), (b) blank solution vs. silver (Ag) nanoparticles in 3-D profiled images and tomographic images, and (c) co-cultured system HUVECs combined with VEGF vs. MG-63 and BMP-2. The geometric illustration of the assembly and fabrication is shown on the left indicating equal aspect ratio, asymmetrical, discontinued configuration, and curved distribution of the two compartments.

In page 7:

.....and the same vapor construction process transformed the assembled iced templates to produce separate compartments of A and B with varied geometries and dimensions in one modulated scaffold. Demonstrations were shown with the versatility to configure compartment A vs. B in equal aspect ratio, asymmetrical, discontinued configuration, and curved distribution of these two compartments.

And

In page 8:

..... Calculated volume ratios for the studied configurations based on analysing the 3D-profiled and micro-CT results were also obtained showing approximately 50% vs. 50% (equal aspect ratio), 56% vs. 44% (asymmetrical), 98% vs. 2% (discontinued configuration), and 26% vs. 74% (curved distribution) for compartment A vs. compartment B, respectively.

o There is no indication of staining in Fig 3c for HUVECS vs MG-63. What is green and what is red? What is cell nuclei and what is growth factor for each component? This is very confusing for readers and detracts from overall understanding of the specific models proposed. Fig S2 does not help provide further clarification as only fluoro VEGF and BMP-2 are shown with no color labels and appear in a different order to those images in Fig 3. These also appear to be different images to Fig 3 which is at odds with the text which describes these images as “merged” images. Fig S2b it is not clear what shapes/examples are trying to be achieved with labelled MG-63 cells. There is no labeling of HUVECS.

Ans:

We appreciate the comments of the reviewer. We have performed addition experiments and reorganized the color channels of the fluorescence signals with respect to each component and specified BMP-2s in green channel, VEGF in red channel, MG-63 in blue channel, and HUVEC in yellow channel. The images of each component with the corresponding fluorescent channel are included in the revised **Figure S4** (due to the addition of new data, the numbering of Figure S2 is now become Figure S4) in the Supplementary Materials, and the merged images were shown in the revised **Figure 3**. In order for the better understanding of readers, indications to

specify these colored signals to the corresponding components are included in the figure captions and also in the Methods section, as suggested by the reviewer. The revised figures are also shown below for the review:

In page 8:

.....Specific staining procedures of each component were carefully performed in each compartment of the same modulated scaffold, as detailed in the Methods. The staining results are summarized in the Supplementary Materials (Figure S4), and the overlaid images in Figure 3c provided overview pictures showing the corresponding shapes and dimensions with defined module boundaries of the multiple cells and biomolecule compartments.

Figure S4. Overview images of varied configurations (with respect to the Figure 3) of a modulated scaffold with compartment A containing the combination of HUVECs/VEGF and compartment B containing MG-63 /BMP-2. (a) Fluorescence images reveal the fabricated modulated scaffolds contain Zip Alexa Fluor™ 488-labelled BMP-2 in green channel, Zip Alexa

Fluor™ 555-labelled VEGF in red channel, PKH26-labelled MG-63 cells in blue channel, and CellTracker™ CMFDA-labelled HUVECs in yellow channel. Cell co-culture was performed at day 1. (b) Additional 3D images were shown to demonstrate an overview of patterned PKH26-labelled MG-63 in three dimensions. Cell co-culture was performed at day 1.

Figure 3. Assembly of various modules with defined chemical/biological components and geometric architecture into one modulated construct with different compartment A vs. B. Components used for assembly and fabrication of the two compartments included (a) Oil Red-O (red channel) vs. fluorescein-5-isothiocyanate (green channel), (b) blank solution vs. silver (Ag) nanoparticles in 3-D profiled images and tomographic images, and (c) co-cultured system HUVECs combined with VEGF vs. MG-63 and BMP-2. The geometric illustration of the assembly and fabrication is shown on the left indicating equal aspect ratio, asymmetrical, discontinued configuration, and curved distribution of the two compartments.

6. There is no information how mass production of scaffold modules was achieved for Fig S3. What are the 10mm cylindrical scaffolds in S3 illustrating?

Ans:

In response to the reviewer's comments, we have included additional images and descriptions for the mass production potential of the scaffold modules in **Figure S5** (due to the addition of new data, the numbering of Figure S3 is now become Figure S5). Briefly, the shape, size, and number of features (mass production) was determined on a PDMS mold (also described in the Methods), which was produced to fabricate the ice templates in (a), and with the proposed fabrication technique, the corresponding scaffold modules were produced in (b). Various shapes and sizes (including 10 mm cylindrical scaffolds) were shown to demonstrate the versatility of scaffold modules that can be fabricated. The revised **Figure S5** is also shown below for the review:

(a) ice templates

vapor fabrication process

(b) scaffold modules

Figure S5. Mass production to fabricate the (a) ice templates and (b) scaffold modules with various shapes and sizes. The shape, size, and number of features was determined on a PDMS mold to produce the ice templates and the vapor fabrication process was then performed to result in the final scaffold modules. The ice templates were produced from solutions containing

blue dyes for the purpose of visualization. The images of (a) and (b) were captured from different batch of samples.

7. Fig 4 pg 13; text pg 7-8 –

o Fig 4 and claims also lack clarity. Fig 4 illustrates 2 compartments A and B for osteogenic and vasculogenic culture from separate MG-63/BMP2 (A) and HUVEC/hASC/VEGF (B) modules. There is no information on the size/shape of the components A and B. Was there a target geometry from one of the examples in Fig 3 that was attempted showing concurrent differentiation and tissue maturation within each of the components together? More critically there is no overview image proving both osteogenic and vascular components shown in Fig 4 have been cultured together as is inferred in text. An image should be provided showing the 2 co-cultured components and indicate the regions of that construct which form the panel of images for compartments A and B in Fig 4a, (similar to examples provided in Fig 3c).

Ans:

In response to the reviewer’s comments, we have included an additional illustration figure in the revised **Figure 4** to better specify the studied size/shape and the linkage to **Figure 3** of the co-cultured component A and B. The images from **Figure 3c** (also **Figure S4** in the Supplementary Materials) indeed indicated the overview of the 2 compartments A and B for osteogenic and vasculogenic culture from separate MG-63/BMP2 in (A) and HUVEC/VEGF in (B), and were shown at the initial stage of co-culture upon assembly the modulated scaffold; however, detailed information was lacking from such overview images, and therefore more details with a relevant scale showing cellular information in terms of cell shape, increased cell population, differentiation and expression activities, along the corresponding differentiation pathway and time scale were thus shown in **Figure 4**. Information regarding the geometry, regions of recorded images on the same scaffold construct is now specified in the illustration and in the captions for **Figure 4**. We have also included an additional discussion to better describe the overview and the linkages between **Figure 3c** and **Figure 4** in the revised manuscript in **page 8**, for the better understanding of readers. The revised figures and discussions are also shown below for the review:

Figure 4. Cell co-cultures of the modulated constructs with defined and customized compartment of A containing the combination of MG-63 and BMP-2, and compartment B containing HUVECs, hASCs, and VEGF. The fabricated scaffold samples with the configuration of an equal aspect ratio for compartment A vs. compartment B was chosen in the study. (a) The recorded immunofluorescence images of fibrous type-I collagen (COL-I) expression at day 7, day 14, and day 21 indicated guided osteogenic activities in compartment-A during the co-cultures. The bone-like nodules formed and increased in size over the indicated period. On the other hand, representative immunofluorescence images of platelet endothelial cell adhesion molecule (PECAM-1/CD31) expression at day 3, day 7, and day 10 showed guided angiogenic activities in compartment B. The classic tube network formed and developed increasingly over the indicated period. (b) Quantitative analysis of osteogenic activities and (c) angiogenic activities of the MG-63 and HUVEC co-culture samples containing compartmentalized BMP-2 and VEGF in compartment A and compartment B, respectively. Statistical analysis was based on three independent experiments with three duplicated samples, and each bar represents the mean value and the standard deviation (\pm SD). Significance levels are indicated according to the unpaired t-test analysis (* $p < 0.05$; and ** $p < 0.01$).

Figure 3.

Figure S4. Overview images of varied configurations (with respect to the Figure 3) of a modulated scaffold with compartment A containing the combination of HUVECs/VEGF and compartment B containing MG-63 /BMP-2. (a) Fluorescence images reveal the fabricated modulated scaffolds contain Zip Alexa Fluor™ 488-labelled BMP-2 in green channel, Zip Alexa Fluor™ 555-labelled VEGF in red channel, PKH26-labelled MG-63 cells in blue channel, and CellTracker™ CMFDA-labelled HUVECs in yellow channel. Cell co-culture was performed at day 1. (b) Additional 3D images were shown to demonstrate an overview of patterned PKH26-labelled MG-63 in three dimensions. Cell co-culture was performed at day 1.

In page 8:

.....With the versatility to produce a scaffold construct composed of the spatial arrangements of specified functional modules with established boundaries between different cell types and microenvironments that was demonstrated in Figure 3c, more detailed cell co-cultures were finally performed with this sophisticated module scaffold, and a demonstration of customizable and programmable biofunctionalities were arranged in the compartmentalized A and B with multiple cell types showing independent cascades of spatial and temporal guidance in such a modulated scaffold.

And

.....and fabricated scaffold samples with the configuration of an equal aspect ratio for compartment A vs. compartment B was chosen for the demonstration.

o No alpha-SMA (smooth muscle actin) staining is provided to show the role of ASCs in maturation of vessel networks in combination with HUVEC (CD31 stained), or other nuclear staining to show localization of ASCs. Images do not show mature networks or lumen claimed.

We appreciate the comments of the reviewer. In response to the reviewer's comments, we have included an additional image in new **Figure S7** in the Supplementary Materials to show the HUVECs formed hollow lumen (CD 31 stained) in combination with the localization of ASCs (nuclear stained). We have also included an additional discussion to better describe the networks formation in the revised manuscript in **page 9**, for the better understanding of readers. The new **Figure S7** and discussions are also shown below for the review:

Figure S7. (a) A representative image showed the hASCs interacted with HUVECs' network as a role of feeder cells during the HUVECs maturation. The nuclei of hASCs were stained in blue channel, while the CD 31 markers of HUVEC were stained in green channel. (b) A magnified image showed the hollow lumen was formed during the maturation of HUVECs.

in page 9

.....A clear network lumen was observed at day 7 and day 10, and the network parameters were measured, including total network length (sum of the lengths of all segments within the 3D network) and network branch points, which were 10.2 ± 1.0 mm and 30.1 ± 3.8 per area of interest (1 mm^2), respectively, over 10 days of culture. Additional data indicated hASCs interacted HUVECs' network as a role of feeder cells during the HUVEC maturation by forming hollow lumen, and these data are included in the Supplementary Materials in Figure S7

o Fig 4c and test pg 8 - I do not believe the authors can claim to have measured “tube network formation” from the images provided or claim “A clear tube lumen started to form at day 7...” The images are not conclusive and Day 10 images look even less “vessel” like than at day 7. 10mm long vessels is extremely long and does not appear accurate – normally vessel length is normalized to surface area. Terminology for quantitative approach taken in Fig 4c should be changed to “network formation” and “network branch points”

Ans:

We appreciate the comments of the reviewer. In response to the reviewer’s comments, we have, in the revised **Figure 4a**, including new data for the Day 10 images to better specify the network and lumen formation. We also, in the revised manuscript in **page 9**, explained the total network length and defined it as the sum of the lengths of all segments measured within the 3D network. In addition, we have changed the terminology in the revised **Figure 4c** and used “network formation” and “network branch points” according to the reviewer’s suggestion. The revised **Figure 4** and discussions are also shown below for the review:

Figure 4. Cell co-cultures of the modulated constructs with defined and customized compartment of A containing the combination of MG-63 and BMP-2, and compartment B containing HUVECs, hASCs, and VEGF. The fabricated scaffold samples with the configuration of an equal aspect ratio for compartment A vs. compartment B was chosen in the study. (a) The recorded immunofluorescence images of fibrous type-I collagen (COL-I) expression at day 7, day 14, and day 21 indicated guided osteogenic activities in compartment-A during the co-cultures. The bone-like nodules formed and increased in size over the indicated period. On the other hand, representative immunofluorescence images of platelet endothelial cell adhesion molecule (PECAM-1/CD31) expression at day 3, day 7, and day 10 showed guided angiogenic activities in compartment B. The classic tube network formed and developed increasingly over the indicated period. (b) Quantitative analysis of osteogenic activities and (c) angiogenic activities of the MG-63 and HUVEC co-culture samples containing compartmentalized BMP-2 and VEGF in compartment A and compartment B, respectively. Statistical analysis was based on three independent experiments with three duplicated samples, and each bar represents the mean value and the standard deviation (\pm SD). Significance levels are indicated according to the unpaired t-test analysis (* $p < 0.05$; and ** $p < 0.01$).

in page 9:

.....A clear network lumen was observed at day 7 and day 10, and the network parameters were measured, including total network length (sum of the lengths of all segments within the 3D network) and network branch points, which were 10.2 ± 1.0 mm and 30.1 ± 3.8 per area of interest (1 mm^2), respectively, over 10 days of culture.

8. Pg 7-8; Methods SI – There is insufficient description of the culture methodology. The only information I could find describing culture conditions and media for all the different cell types for osteogenic, vascular and neural tissue formation was in SI pg 4: “The culturing conditions and media for cell culture were used according to the reported methods as referenced.” It is not clear which references refer to this media and there is insufficient information for readers to repeat the methodological steps in this specific study.

Ans:

We appreciate the comments of the reviewer. We have, in the Supplementary Materials in **page 3**, additionally included more sufficient information of the culture methodology. The additional information is also shown below for the review:

in Supplementary Materials, page 3, Methods:

.....hASCs, MC3T3-E1, and MG-63 cells were cultured in MEM-alpha (Thermo Fisher Scientific, USA) supplemented with 10% fetal bovine serum (FBS, Biological Industries, Israel) and 1% antibiotic/antimycotic solution (Biological Industries, Israel). PC 12 cells were cultured in RPMI-1640 medium (Thermo Fisher Scientific, USA) supplemented with 5% FBS (Biological Industries, Israel), 10% heat-inactivated horse serum (Biological Industries, Israel), and 1% antibiotic/antimycotic solution (Biological Industries, Israel). HUVECs were cultured in M199 medium (Sigma-Aldrich, USA) supplemented with 25 U/ml heparin (Sigma-Aldrich, USA), 30 µg/ml endothelial cell growth supplement (ECGS, Sigma-Aldrich, USA), 10% FBS (Biological Industries, Israel), and 1% antibiotic/antimycotic solution (Biological Industries, Israel). Culturing conditions were maintained at 37°C with a humidified atmosphere containing 5% CO₂ and 95% air, and media were changed once in every 3-4 days. The cell loading density was approximately 1×10^5 cells per sample during the experiments unless otherwise indicated, and the fabricated cell-laden scaffold samples were cultured immediately in the above-mentioned corresponding growth medium, or induction medium to identify osteogenesis and neurogenicity.

o Furthermore in line with point 7 and point 8 above, co-culture of multiple tissue types requires complex combinations of individual media components (e.g. osteogenic and vasculogenic media) in order to support appropriate long-term (2-3 week) culture of merged tissue modules. There is no description of media to achieve examples Fig 4 for both combined osteogenic and vasculogenic differentiation over d10-d21. The fact that the compartments A and B were cultured

for d21 and d10 each respectively seems to suggest that they were not co-cultured together as inferred. Further clarity and data is required to prove the claims in the study.

Ans:

We appreciate the comments of the reviewer. We have, in the Supplementary Materials in **page 4**, additionally included more sufficient information of the culture media for both combined osteogenic and vasculogenic differentiation over day 10 to day 21. Images taken from the compartments A and B are based on the time required for osteogenic and vasculogenic differentiation [Maehata et al., *Journal of Oral Biosciences* 2009, 51, 72; Bird et al., *Journal of Cell Science* 1999, 112, 1989]. The independent and distinct biological activities detected in compartments A and B resulted from the cell differentiations in the corresponding compartments, which supported the hypothesis. We have also, in the revised manuscript in **page 9**, explained why the images from compartments A and B were taken in different time course. The additional information and clarity are also shown below for the review:

in Supplementary Materials, page 4:

.....The co-cultural samples were incubated with a culture medium that combined equal volume of MEM-alpha (Thermo Fisher Scientific, USA) and M199 medium (Sigma-Aldrich, USA) and supplemented with 10% FBS (Biological Industries, Israel), 1% antibiotic-antimycotic (Biological Industries, Israel), 10×10^{-3} M β -glycerol phosphate (Sigma-Aldrich, USA), 100 μ g/mL L-ascorbic acid (Sigma-Aldrich, USA), 0.1×10^{-6} M dexamethasone (Sigma-Aldrich, USA), 25 U/ml heparin (Sigma-Aldrich, USA), 30 μ g/ml ECGS (Sigma-Aldrich, USA) for further characterization.

in main manuscript, page 9:

.....The expression of both markers over time was evaluated based on the differentiation pathways of osteogenesis and angiogenesis; the detected immunofluorescence signals showed an enhancement in COL-I expression of MG-63 cells from day 7 to day 21 corresponding to osteogenesis activity in compartment A and enhanced CD31 signals in vascularized HUVEC cells from day 3 to day 10 for potential angiogenesis activity in B. The independent and distinct biological activities detected in compartments A and B resulted from the cell differentiations in the corresponding compartments, which supported the hypothesis.

9. There is no discussion on how this proposed fabrication approach compares with other existing modular assembly approaches or describes drawbacks of the proposed strategy to put

this work into context for readers. What is the benefit of this vapor phased approach to fabricate modular tissues offer compared to existing approaches for fabrication of complex 3D hierarchical tissues? The results of this study have relatively low impact given the previous work in 3D bioprinting of multi-material hydrogel bioinks. Previous examples including bottom-up assembly of preformed cell laden hydrogels (e.g. from Khademhosseini, Zhang) or 3D Bioprinting/biofabrication methods containing multimaterial bioinks and multicellular bioinks/modules/spheroids (e.g. Forgacs; Nakayama; Lutolf; Lewis).

Ans:

We appreciate the comments of the reviewer. We agree with the reviewer that the existing hydrogel bioinks and/or 3D printing did show some advantages for the fabrication of scaffolding materials with capability to multiple materials and functions, and certainly there are more other techniques available including extrusion prototyping, laser sintering, direct assembly, aggregation, and etc. Advantages and disadvantages can be seen from one technique when compare to another. For instance, the hydrogels technique usually suffers problems of being weak in mechanical properties even after being cross-linked (natural hydrogels), poor biocompatibility with producing toxic non-natural degradation products and lack of bioactive ligands (synthetic hydrogels), and difficulty to control the porous structures [Nicodemus et al., *Tissue Engineering Part B: Reviews* 2008, 14, 149-165; Nichol et al., *Biomaterials* 2010, 31, 5536-5544; Mao et al., *Progress in Natural Science: Materials International* 2020, 30, 618; Li et al., *Materials & Design* 2018, 159, 20]. Complex bioinks were formulated to accommodate a wide range of applications, but additional energy sources or chemicals were involved, and special modulation equipment was required for the fabrication [Billiet et al., *Biomaterials* 2014, 35, 49-62; Khalil et al., *Journal of Biomechanical Engineering* 2009, 131, 111002; Derakhshanfar et al., *Bioactive Materials* 2018, 3, 144]. Nevertheless, the current existing methods were performed by addition or subtraction materials (in a desired control volume) to construct a (porous) scaffold material, but the current study, with a different prospect of view, provided a method that the scaffold construction is produced by exchanging materials (sublimating ice template and depositing polymer) in the same control volume. A rationale was shown to accommodate multiple and distinct functional molecules including different types of cells during the fabrication process with versatility, and these accommodations and fabrications are also modulated with facility. The novelty of the introduced vapor fabrication method, and the above-mentioned versatility and facility are emphasized, advantages of the current method compared to existing methods were also summarized in the revised manuscript in **page 2** and **3**. References suggested by the reviewer are cited [ref. 7-10]. The additional discussions are also shown below for the review:

In page 2:

.....Current methods to apply module assemblies include direct assembly and/or aggregation, cell-laden hydrogels, cell sheets or direct 2D or 3D printing techniques and are reported and reviewed elsewhere [ref. 3-10].

and

.....With these stringent fabrication requirements, only sporadic methods including cytocompatible hydrogels, bioinks for printing or spinning from 2D to 3D, and/or similar approaches are available thus far. The dilemma, however, is that hydrogels provide excellent cellular cytocompatibility due to their hydrated material properties but usually lack overall high mechanical integrity, and exist potential toxic degradation products. Complex bioinks were formulated to accommodate a wide range of applications, but additional energy sources or chemicals were involved, and special modulation equipment was required for the fabrication. Ideal modular fabrication has yet to be achieved,⁶ and its combination with biological complexity may pave the way for the development of a new generation of scaffolds.

In page 3:

.....The fabricated scaffold modules offered the advantages of (I) a straightforward accommodation of chemical/biological composition with multiple components ranging from functional biomolecules to living cells with determined composition ratio and customizable combination of these components, (II) a benign vapor-phase fabrication process, which utilizes ice/water templates for vapor sublimation and a USP (United States Pharmacopeia) Class VI highly biocompatible poly-p-xylylene for vapor deposition, (III) controlled mass transport of species in a defined construction volume to avoid phase separation of the components, (IV) connected pore structure formation with tuneable mechanical properties allowing for interaction between the preloaded components and cells, and (V) a robust discontinued process of assembling modules during the templating stage, followed by a continued, one-step vapor deposition process resulting in a continuous scaffold construct composed of the spatial arrangements of specified functional modules with established boundaries between different cell types and microenvironments.

Response to Reviewer 3:

Comments:

The authors fabricated 3D porous scaffolds with new functionalities by the addition of multiple dopants using a home-built sublimation/deposition system. They then tested their constructs for tissue engineering with multiple cells. In general, the discussion and results are well addressed, and the conclusions well supported by the results obtained. However, some issues, suggestions and curiosities are pointed out that might improve the quality of the manuscript:

We appreciate the comments of the reviewer and are very delighted with the positive feedback.

1. The authors state “the iced template was formed by evaporating the water molecules, instead of depositing poly-p-xylylene polymer to replace the evaporated volume of water and forming a porous scaffold”, which might lead to confusion. How is it the polymer controlled to be deposited at the same time is sublimated without been replaced? Is the speed of deposition an asset?

Ans:

We thank the reviewer to point out the grammar issue and the typo. The correct sentence and description is now included in the revised manuscript in **page 4**, and is also shown below for the review:

In page 4:

.....the iced template was sublimated with evaporation of water molecules, and a depositing poly-p-xylylene polymer occurred to replace the evaporated volume of water and forming a porous scaffold containing the preloaded FGF-2 and hASCs.

2. How long is the fabrication process from introduction of cells until incubation?

Ans:

The reviewer raises an excellent point regarding the fabrication time. According to our previous report on the vapor sublimation and disposition mechanism [Tung et al., *Applied Materials Today* 2017, 7, 77; Tung et al., *Nature Communications* 2018, 9, 2564], the volume size of the scaffolding product fabrication is proportional to the processing time, and briefly it takes approximately 60 mins for a 5 cm³-sized sample. To ensure cell viability for cell-laden samples, once the scaffold fabrication is done, immersing the samples in the culture medium is performed immediately upon retrieving the samples from the deposition chamber. We have also in the revised manuscript in **page 4**, included a discussion of the processing time and handling the cell-laden samples, as suggested by the reviewer. The additional discussion is also shown below for the review:

In page 4:

.....The fabrication is a time-dependant process to produce a proportional volume of scaffold product and required approximately 60 mins for a 5 cm³-sized sample. To ensure cell viability for cell-laden samples, immersing the samples in the culture medium was performed immediately upon retrieving the samples from the deposition chamber.

3. Please specify the color code in the immunofluorescence images.

Ans:

We appreciate the comments of the reviewer. We have, in the revised manuscript in **Figure 3** and the **Figure S4** (in the Supplementary Materials), specified the color code for better indications of the immunofluorescence images, as suggested by the reviewer. The revised descriptions and figures are also shown below for the review:

In page 8:

.....Specific staining procedures of each component were carefully performed in each compartment of the same modulated scaffold, as detailed in the Methods. The staining results are summarized in the Supplementary Materials (Figure S4), and the overlaid images in Figure 3c provided overview pictures showing the corresponding shapes and dimensions with defined module boundaries of the multiple cells and biomolecule compartments.

Figure S4. Overview images of varied configurations (with respect to the Figure 3) of a modulated scaffold with compartment A containing the combination of HUVECs/VEGF and compartment B containing MG-63 /BMP-2. (a) Fluorescence images reveal the fabricated modulated scaffolds contain Zip Alexa Fluor™ 488-labelled BMP-2 in green channel, Zip Alexa Fluor™ 555-labelled VEGF in red channel, PKH26-labelled MG-63 cells in blue channel, and CellTracker™ CMFDA-labelled HUVECs in yellow channel. Cell co-culture was performed at day 1. (b) Additional 3D images were shown to demonstrate an overview of patterned PKH26-labelled MG-63 in three dimensions. Cell co-culture was performed at day 1.

Figure 3. Assembly of various modules with defined chemical/biological components and geometric architecture into one modulated construct with different compartment A vs. B. Components used for assembly and fabrication of the two compartments included (a) Oil Red-O (red channel) vs. fluorescein-5-isothiocyanate (green channel), (b) blank solution vs. silver (Ag) nanoparticles in 3-D profiled images and tomographic images, and (c) co-cultured system HUVECs combined with VEGF vs. MG-63 and BMP-2. The geometric illustration of the assembly and fabrication is shown on the left indicating equal aspect ratio, asymmetrical, discontinued configuration, and curved distribution of the two compartments.

4. Please add P-value and the number of repeats done for each experiment below the histograms to show the statistical significance differences and consistency of your results.

Ans:

We appreciate the comments of the reviewer. We have, in the revised manuscript, included statistical significance indications for all the relevant data including **Figure 2a** (FGF v control), **Figure 2b** (effect of BMP-2 on OCN, ALP, Ca v control), **Figure 2c** PEDOT/PSS/PRP variants versus control for neurite length or neurogenic activity, **Figure 4b** (col I, nodule formation d7-21), **Figure 4c** (tube formation, branching d3-d10), and **Figure S6** (due to the addition of new data, the numbering of Figure S4b is now become Figure S6b), as suggested by the reviewer, and also for the better understanding of readers. The revised **Figures** of data are also shown below for the review:

Figure 2. Fabrication of various modules with specified cell-guiding activities. (a) SEM and laser confocal images were recorded for modules containing hASCs cells and FGF-2 and showed cell

proliferation activities after culturing for 4 days. The F-actin cytoskeleton (green) was stained by Alexa Fluor® 488-conjugated phalloidin and the nucleus (red) was stained with propidium iodide; the inset images showed attached and spreaded cells. (b) Modules composed of MC3T3-E1 cells and BMP-2 showed guided osteogenesis activities. The ALP expression was analyzed after 7 days of culture, while OCN expression and calcium deposition were analyzed after 21 days of culture. (c) Modules composed of more complex components, including PC12 cells, PRP molecules, and the PEDOT:PSS system, were fabricated and showed guided neurogenesis activities after 5 days of culture. Data are expressed as the mean value with the standard deviation based on three independent samples. Significance levels are indicated according to the unpaired t-test (n.s., nonsignificant difference; * $p < 0.05$; ** $p < 0.01$; and *** $p < 0.001$).

Figure 4. Cell co-cultures of the modulated constructs with defined and customized compartment of A containing the combination of MG-63 and BMP-2, and compartment B containing HUVECs, hASCs, and VEGF. The fabricated scaffold samples with the configuration of an equal aspect ratio for compartment A vs. compartment B was chosen in the study. (a) The recorded immunofluorescence images of fibrous type-I collagen (COL-I) expression at day 7, day 14, and day 21 indicated guided osteogenic activities in compartment-A during the co-cultures. The bone-like nodules formed and increased in size over the indicated period. On the other hand, representative immunofluorescence images of platelet endothelial cell adhesion molecule (PECAM-1/CD31) expression at day 3, day 7, and day 10 showed guided angiogenic activities in compartment B. The classic tube network formed and developed increasingly over the indicated period. (b) Quantitative analysis of osteogenic activities and (c) angiogenic activities of the MG-63 and HUVEC co-culture samples containing compartmentalized BMP-2 and VEGF in compartment A and compartment B, respectively. Statistical analysis was based on three independent experiments with three duplicated samples, and each bar represents the mean value and the standard deviation (\pm SD). Significance levels are indicated according to the unpaired t-test analysis (* $p < 0.05$; and ** $p < 0.01$).

Figure S6. (a) Immunofluorescence images show the specific expression of the protein markers on MG-63 (human osteoblasts) and HUVECs (human umbilical vein endothelial cells). COL-I, type-I collagen; OCN, osteocalcin; CD31, cluster of differentiation 31 (platelet endothelial cell adhesion molecule); and VEGFR2, vascular endothelial growth factor receptor-2. (b) Quantitative analysis of the protein marker expression on MG-63 and HUVECs not only confirmed the cell identities but also verified the methodology to visualize MG-63 and HUVECs specifically in the fabricated modulated scaffolds. Statistical results were analysed based on three independent experiments of three duplicated samples, and each bar represents the mean value and the standard deviation (\pm SD). The significance level is shown by the unpaired t-test. *** $p < 0.001$.

5. Please include controls of the culture of hASCs (Figure 2a), so the stated “enhanced attaching, spreading, and growth of hASCs on the porous structures” can be visualized.

Ans:

We appreciate the comments of the reviewer. We have performed additional control experiments as suggested by the reviewer. These additional data include images from SEM and confocal microscope results to compare the FGF-2-decorated samples with samples without FGF-2, and are shown in the revised **Figure 2**, and the cell numbers are also statistically compared. Furthermore, additional Live/Dead staining images are included in the Supplementary Materials in **Figure S2** to indicate the cell population were equivalent between the FGF-2-decorated samples and control samples at day 1, while higher cell number was found for the FGF-2-decorated group at day 4. The revised **Figure 2**, additional discussions, and **Figure S2** are also shown below for the review:

Figure 2. Fabrication of various modules with specified cell-guiding activities. (a) SEM and laser confocal images were recorded for modules containing hASCs cells and FGF-2 and showed cell

proliferation activities after culturing for 4 days. The F-actin cytoskeleton (green) was stained by Alexa Fluor® 488-conjugated phalloidin and the nucleus (red) was stained with propidium iodide; the inset images showed attached and spreaded cells. (b) Modules composed of MC3T3-E1 cells and BMP-2 showed guided osteogenesis activities. The ALP expression was analyzed after 7 days of culture, while OCN expression and calcium deposition were analyzed after 21 days of culture. (c) Modules composed of more complex components, including PC12 cells, PRP molecules, and the PEDOT:PSS system, were fabricated and showed guided neurogenesis activities after 5 days of culture. Data are expressed as the mean value with the standard deviation based on three independent samples. Significance levels are indicated according to the unpaired t-test (n.s., nonsignificant difference; * $p < 0.05$; ** $p < 0.01$; and *** $p < 0.001$).

In page 5:

..... Additional control experiments were performed by Live/Dead staining showing the comparison of the FGF-2-decorated samples with samples without FGF-2 at day 1 and day 4, and the data are included in the Supplementary Materials in Figure S2.

Figure S2. (a) Fluorescence images by Live/Dead staining comparing the FGF-2-decorated samples and the control group of samples (without FGF-2). (b) Comparison of the calculated live cell number revealing the cell population were equivalent between the FGF-2-decorated samples and control samples at day 1, while higher cell number (** $p < 0.01$) was found for the FGF-2-decorated group at day 4. (c) Cell viability analysis based on the ratio of total viable

cells/total cells revealed both FGF-2-decorated and control samples have a good biocompatibility with no statistically significant difference. Data were analysed based on three independent experiments of three duplicated samples, and each bar represents the mean value and the standard deviation (\pm SD). Significance levels are shown by the unpaired t-test. n.s., nonsignificant difference.

6. Related to the “spreading and growth” of previous statement, according to Figure 2a, hASCs cells after 4 days of culture have the same roundish shape as in day 1, typical of unfold not extended cells, and no actin filaments can be distinguished. That can denote no functionality of the cells, i. e., no effective scaffolds for these cells. Please provide evidence of spreading, functionality and growth with other more specific function/proliferation markers, zoom-in images, etc.

Ans:

We appreciate the comments of the reviewer. We have, in the revised manuscript in **Figure 2a**, presented higher resolution images to show the spreading and growing cells on scaffolds, and additional high magnification images are also provided in the inset to indicate actin/phalloidin and PI staining for the better understanding of readers, as suggested by the reviewer. The modified **Figure 2** are also shown below for the review:

Figure 2. Fabrication of various modules with specified cell-guiding activities. (a) SEM and laser confocal images were recorded for modules containing hASCs cells and FGF-2 and showed cell proliferation activities after culturing for 4 days. The F-actin cytoskeleton (green) was stained by Alexa Fluor® 488-conjugated phalloidin and the nucleus (red) was stained with propidium iodide; the inset images showed attached and spreaded cells. (b) Modules composed of MC3T3-E1 cells and BMP-2 showed guided osteogenesis activities. The ALP expression was analyzed after 7 days of culture, while OCN expression and calcium deposition were analyzed after 21 days of culture. (c) Modules composed of more complex components, including PC12 cells, PRP molecules, and the PEDOT:PSS system, were fabricated and showed guided neurogenesis activities after 5 days of culture. Data are expressed as the mean value with the standard deviation based on three independent samples. Significance levels are indicated according to the unpaired t-test (n.s., nonsignificant difference; * $p < 0.05$; ** $p < 0.01$; and *** $p < 0.001$).

7. What does “cell activity” in Figure 2a refer to? How is it calculated? Usually “activity” is synonym of “function”, but according to the manuscript only an MTT assay has been done. May the authors have referred to viability instead? Please clarify.

Ans:

We appreciate the comments of the reviewer. We agree with the reviewer that MTT assay data usually refer to as the cell viability, or cell metabolic activity [Berridge et al. *Biochemica* 1996, 4,14-19; Berridge et al. *Biotechnology Annual Review* 2005, 11, 127-152]. However, the MTT data was obtained from the same batch of viable cells imaged in **Figure 2a** and from the same studied periods of their proliferation activities, we therefore use the term “metabolic activity” here to avoid confusion with the Live/Dead data (which was usually termed cell viability) that was mentioned in previous section of the manuscript. The modified **Figure 2** is also shown below for the review:

Figure 2. Fabrication of various modules with specified cell-guiding activities. (a) SEM and laser confocal images were recorded for modules containing hASCs cells and FGF-2 and showed cell proliferation activities after culturing for 4 days. The F-actin cytoskeleton (green) was stained by Alexa Fluor® 488-conjugated phalloidin and the nucleus (red) was stained with propidium iodide; the inset images showed attached and spreaded cells. (b) Modules composed of MC3T3-E1 cells and BMP-2 showed guided osteogenesis activities. The ALP expression was analyzed after 7 days of culture, while OCN expression and calcium deposition were analyzed after 21 days of culture. (c) Modules composed of more complex components, including PC12 cells, PRP molecules, and the PEDOT:PSS system, were fabricated and showed guided neurogenesis activities after 5 days of culture. Data are expressed as the mean value with the standard deviation based on three independent samples. Significance levels are indicated according to the unpaired t-test (n.s., nonsignificant difference; * $p < 0.05$; ** $p < 0.01$; and *** $p < 0.001$).

8. Why ALP was evaluated after 7div while OCN analyses lasted 21 days? How many days of culture for the calcium analyses? Please include the images of ALP and calcium.

Ans:

In response to the reviewer's comments, the studied periods to characterize osteogenesis activities (e.g. ALP and calcium) are based on the reported results showing the required time of culture that these activities can then be detected. ALP activity is involved in the early stage of osteogenic differentiation while mineralization of calcium deposition is involved in the late stage [Zhang et al., *International Journal of Oral Science*, 2019, 11, 12]. We therefore chose to examine the ALP activity after 7 days of culture, while perform OCN analyses and calcium analysis were performed after 21 days of culture. The osteogenesis activities, determined by OCN, ALP, and calcium analyses as shown in **Figure 2b** should be considered as complementary evidence to show that BMP-2-conditioned samples exhibited better osteogenic guidance toward MC3T3-E1 cells than the control group of samples. The ALP activities and calcium depositions were directly semi-quantified via colorimetric determination using an ELISA reader as the protocols described previously [Wu et al., *ACS Applied Bio Materials* 2020, 3, 7193], so that no images were obtained. We have, in the revised Supplementary Materials, added details of the related information including the studied period in **page 4** (Methods), as suggested by the reviewer. Relevant references are also cited. This additional information is also shown below for the review:

in Supplementary Materials, page 4, Methods:

..... In order to determine osteogenic activity, the ALP activity of the scaffold containing MC-3T3-E1 cells was analysed after 7 days of culture, and OCN expression and calcium deposition were analysed after 21 days of culture.

9. How many days were PC12 cultured in the scaffolds (Figure 2c)? Please include all the information related, including seeding density and media type and changes.

Ans:

We appreciate the comments of the reviewer. We have, in the revised Supplementary Materials, added detailed information including the culture period, seeding density, media types, and changes in **page 3-4** (Methods Section), as suggested by the reviewer. This additional information is also shown below for the review:

in Supplementary Materials, page 3-4, Methods:

.....PC 12 cells were cultured in RPMI-1640 medium (Thermo Fisher Scientific, USA) supplemented with 5% FBS (Biological Industries, Israel), 10% heat-inactivated horse serum (Biological Industries, Israel), and 1% antibiotic/antimycotic solution (Biological Industries, Israel).

.....Culturing conditions were maintained at 37°C with a humidified atmosphere containing 5% CO₂ and 95% air, and media were changed once in every 3-4 days. The cell loading density was approximately 1 × 10⁵ cells per sample during the experiments unless otherwise indicated, and the fabricated cell-laden scaffold samples were cultured immediately in the above-mentioned corresponding growth medium, or induction medium to identify osteogenesis and neurogenicity.

.....To determine neurogenesis activity, after 5 days of culture, samples containing PC 12 cells were stained by immunofluorescence for the neuronal markers, Nestin and Tuj-1.

10. How does the doping of FGF-2, BMP-2, PRP, and PEDOT:PSS affects the scaffold's mechanical properties? It is strongly suggested to provide analyses of, at least, the Young's Modulus and swelling.

Ans:

We appreciate the comments of the reviewer. In our previous studies, dopants were tried including inorganic Au, Ag, Fe oxides, carbon dioxide, and also organic solvents, such as methanol, ethanol, acetone, hexane. Briefly, these dopants were classified (by us) into two categories of "volatile" and "non-volatile" during the processing thermodynamic conditions (approximately 0.1 bar and room temperature or below), and we found the fabricated porous polymer products were greatly impacted with their mechanical properties, e.g. pore size, porosity, Young's modulus, and etc., by "volatile" dopants and was found due to the sublimation (volatility)

during the fabrication process [Tung et al., *Applied Materials Today* 2017, 7, 77; Chiu et al., *Chemistry of Materials* 2020, 32, 1120], and these mechanical properties were tunable to show in a wide range, for instance, pore sizes ranging from $>5\ \mu\text{m}$ to $100\ \mu\text{m}$, and porosities of approximately 50% - 80%, Young's modulus from $\sim 100\ \text{kPa}$ to $10000\ \text{kPa}$, were attempted and tuned. By contrast, the "non-volatile" dopants were showing negligible impact to the mechanical properties of fabricated porous products, due to (i) non-sublimating behavior, and (ii) small amount of dopants compare to the polymer matrix; and the overall mechanical properties were assume consistent to a blank porous product (only with polymer matrix). Considering the dopants used in the current study, e.g. FGF-2, BMP-2, PRP, and PEDOT:PSS were in the "non-volatile" category and were used in relatively small amount compared to the polymer scaffold matrix, consistent and negligible impact of these dopants to the overall mechanical properties are expected. Additional measurements of these properties were performed and were discussed in the revised manuscript in **page 2-3** and **page 4**, as suggested by the reviewer. The additional discussions are also shown below for the review:

In page 2-3:

.....With using volatile compounds as dopants, the mechanical properties of the resultant porous materials were tuneable in a wide range, for instance, pore sizes ranging from $>5\ \mu\text{m}$ to $100\ \mu\text{m}$, and porosities of approximately 50% - 80%, Young's modulus from $\sim 10\ \text{kPa}$ to $10000\ \text{kPa}$, were attempted and tuned; while the use of non-volatile dopants render a uniform or controlled localization of encapsulating the dopants and without interference to the mechanical properties.

And

In page 4:

..... With also tuneable mechanical properties to fabricate the scaffold modules,²¹ consistent properties including approximately $35.7 \pm 8.2\ \mu\text{m}$ in pore size, $63.4\% \pm 6.3$ porosity, and $150 \pm 21.5\ \text{kPa}$ for the Young's modulus, were measured and used in the current studies.

11. Why

Ans:

We believe this might be a typo by the reviewer.

12. Labels (i), (ii), (iii) have been used several times along the manuscript. What do these labels at lines 214-216 refer to? As well, which is compartment A and B in the scaffold's structure? A schematic representation and inclusion of labels in Figure 3 will be appreciated to facilitate understanding.

Ans:

In response to the reviewer's comments, these labels (i), (ii), (iii) used here at line 214-216 (**page 7**) exploited the same modulation concept from **page 5**, and shared the same descriptions to demonstrate the versatility of applying different biological components. We have also, in the revised manuscript in **page 7**, emphasized the relations of these labels (i), (ii), (iii) that were linked to the descriptions in **page 5** for the better understanding of readers. Plus, we have corrected the use of numbering system in **page 3**, showing (I), (II), (III), (IV)...for describing the advantages of our methods and to avoid the confusion. The compartment A and B denote to different compartments in one modulated and fabricated scaffold, and we have also included additional indications to better define the compartment A and B in **page 7-8** and in the captions of **Figure 3**, as suggested by the reviewer. The addition changes and the revised **Figure 3** are also shown below for the review:

In page 7:

.....and the same vapor construction process transformed the assembled iced templates to produce separate compartments of A and B with varied geometries and dimensions in one modulated scaffold. Demonstrations were shown with the versatility to configure compartment A vs. B in equal aspect ratio, asymmetrical, discontinued configuration, and curved distribution of these two compartments.

In page 8:

..... Calculated volume ratios for the studied configurations based on analysing the 3D-profiled and micro-CT results were also obtained showing approximately 50% vs. 50% (equal aspect ratio), 56% vs. 44% (asymmetrical), 98% vs. 2% (discontinued configuration), and 26% vs. 74% (curved distribution) for compartment A vs. compartment B, respectively.

Figure 3. Assembly of various modules with defined chemical/biological components and geometric architecture into one modulated construct with different compartments of A and B. Components used for assembly and fabrication included (a) Oil Red-O (red channel) vs. fluorescein-5-isothiocyanate (green channel), (b) silver (Ag) nanoparticles vs. blank solution in 3-D profiled images and tomographic images, and (c) HUVECs combined with VEGF vs. MG-63 and BMP-2. The geometric illustration of the assembly and fabrication is shown on the left indicating equal aspect ratio, asymmetrical, discontinued configuration, and curved distribution of the two compartments.

13. Which is the final aim of the co-cultures in separate compartments? And why have the authors chosen those specific cell lines?

Ans:

In response to the reviewer's comments, the co-culture experiments intended to demonstrate a modulated construct of interest assembled from separate and distinct biological systems (compartmentalized) that comprised of varied cell lines, growth factors, and functional molecules were cultured accordingly in the same culture conditions and period, and the compartmentalized systems were developed independently along the devised and expected pathways. The chosen cell lines including bone-forming and endothelial cell models intended to demonstrate as an example in light of the recognized histological observation of human bone tissues, where osteoblasts and osteoprogenitor cells are located adjacent to blood vessel endothelial cells [Villars et al., *J. Cell Biochem.* 2000, 79, 672; Kargozar et al., *Applied Sciences* 2019, 9, 174]. We have also, in the revised manuscript in **page 8**, included additional discussions to emphasize these matters, as suggested by the reviewer. The additional discussions are also shown below for the review:

In page 8:

.....Cell co-cultures were finally performed with this sophisticated module scaffold, and a demonstration of customizable and programmable biofunctionalities were arranged in the compartmentalized A and B with multiple cell types showing independent cascades of spatial and temporal guidance in such a modulated scaffold.

and

.....The combination was chosen in light of the recognized histological observation of human bone tissues, where osteoblasts and osteoprogenitor cells are located adjacent to blood vessel endothelial cells.

Reviewers' Comments:

Reviewer #1:

Remarks to the Author:

The authors have answered my comments sufficiently and have added new data into their manuscript.

Reviewer #2:

Remarks to the Author:

The authors present a revised manuscript (NCOMMS-20-42730A) entitled "Vapor-Phased Fabrication and Modulation of Cell-Laden Scaffolding Materials".

The manuscript has been significantly improved with new data and in particular improved clarity in Figures and new control samples and statistical analysis addressing original conclusions. Additional information on characterization of each step of the vapor-phased process has been provided.

I still have remaining concerns around how this vapor-phased process provides improvements or new novel knowledge over existing fabrication approaches - particularly hybrid biofabrication approaches - which allow targeted and automated hierarchical control of cell placement and ECM cues in advanced bioinks at a resolution and with shape fidelity and multicomponent physical and bioactive cues that address technological challenges proposed in this vapor-phased approach but with more simplistic component A and B examples provided.

Reviewer #3:

Remarks to the Author:

The authors have responded to my comments and criticism. The manuscript can now be accepted in Nature Communications.

We would like to thank all reviewers of our manuscript for their comments and suggestions for improvement on our manuscript. In the following, we will address all comments and explain our rational and resulting changes to the manuscript in detail. The original statements of the reviewers are shown in plain black, our responses in bold red. For easiness of follow-up, we marked the changes yellow in the main manuscript.

Response to Reviewer 2:

Comments:

The authors present a revised manuscript (NCOMMS-20-42730A) entitled “Vapor-Phased Fabrication and Modulation of Cell-Laden Scaffolding Materials”.

The manuscript has been significantly improved with new data and in particular improved clarity in Figures and new control samples and statistical analysis addressing original conclusions. Additional information on characterization of each step of the vapor-phased process has been provided.

We appreciate the comments of the reviewer and are very delighted with the positive feedback.

I still have remaining concerns around how this vapor-phased process provides improvements or new novel knowledge over existing fabrication approaches - particularly hybrid biofabrication approaches - which allow targeted and automated hierarchical control of cell placement and ECM cues in advanced bioinks at a resolution and with shape fidelity and multicomponent physical and bioactive cues that address technological challenges proposed in this vapor-phased approach but with more simplistic component A and B examples provided.

Ans:

We appreciate the comments of the reviewer. In response to the reviewer’s comments, we agree with the reviewer that the current bioinks printing technique is advantageous with the automated hierarchical control to achieve complex structure and with high shape and dimension fidelity, however:

1. The bioinks used are in a liquid phase, which the phase separation problem between molecule/molecule and/or molecule/medium is usually unavoidable during the extrusion and printing process due to the molecular mass transport phenomenon intends to reach a steady-state, and the problem is difficult to control [Gokmen et al., *Prog. Polym. Sci.* 2012, 37, 365].

→ the current vapor-phased fabrication process used an iced template, which the control of mass transport phenomenon is designed being performed at an unsteady-state (for sure being

able to perform at a steady-state) to avoid a phase separation between the accommodated multiple molecules and cells.

2. The bioinks used are subjected to shear stress, extrusion, injection, which can cause irritations to the sensitive biomolecules and living cells [Derakhshanfar et al., *Bioactive Materials* 2018, 3, 144; Wu et al., *ACS Appl. Mater. Interfaces* 2019, 11, 22152; Naghieh et al., *Journal of Pharmaceutical Analysis* 2021, DOI: <https://doi.org/10.1016/j.jpha.2021.02.001>], and the final formation of the matrix (e.g. hydrogels) requires an additional crosslinking process which also involved potentially irritations including solvents, intensive energies of by thermo or UV lights [Murphy et al., *Nature Biotechnology* 2014, 32, 773; Hennink et al., *Advanced Drug Delivery Reviews* 2012, 64, 223].

→ the current reported technique, exhibited no mechanical irritations to these sensitive molecules and cells, and the formation of the matrix occurred, again, by controlling the mass transport of the vapor sublimation of the ice templates (water solvent/medium) and the vapor deposition of polymer molecules to replace the ice template, forming the final matrix of a scaffold. It is a vapor-phased molecular exchange process due to mass transport and exerts no irritation energies compared to the existing bioink/printing techniques.

We have therefore in the revised manuscript, emphasized these advantages and comparisons of our proposed method compared to the existing techniques (including bioink/printing and others) with additional discussions in **page 2**, and **3** for the better understanding of readers. Relevant references are also cited. The additional discussions are also shown below for the review:

In page 2:

.....With these stringent fabrication requirements, only sporadic methods including cytocompatible hydrogels, bioinks for printing or spinning from 2D to 3D, and/or similar approaches are available thus far. The challenges, however, are that hydrogels provide excellent cellular cytocompatibility due to their hydrated material properties but usually lack overall high mechanical integrity, and exist potential toxic degradation products; the bioinks printing techniques, on the other hand, although provide excellent automated hierarchical control and with high shape fidelity, and are formulated to accommodate a wide range of applications, sensitive molecules and cell suffer irritations by shear stresses during extrusion and injection, by additional energy sources or chemicals are involved, special modulation equipment is required for the fabrication, and an unavoidable phase separation and dislocation of biomolecules and/or laden cells occurs due to mass transport in such liquid-phased bioinks.

In page 3:

.....The fabricated scaffold modules offer the advantages of (I) a straightforward accommodation of chemical/biological composition with multiple components ranging from functional biomolecules to living cells with determined composition ratio and customizable combination of

these components, (II) a benign vapor-phase fabrication process, which utilizes ice/water templates for vapor sublimation and a USP (United States Pharmacopeia) Class VI highly biocompatible poly-p-xylylene for vapor deposition, forming a scaffold matrix without irritating sensitive molecules and cells, (III) control mass transport of species capable of being operated at an unsteady-state and/or steady-state conditions in a defined construction volume to avoid phase separation and dislocation of the components, (IV) connect pore structure formation with tuneable mechanical properties allowing for interaction between the preloaded components and cells, and (V) a robust discontinued process of assembling modules during the templating stage, followed by a continued, one-step vapor deposition process resulting in a continuous scaffold construct composed of the spatial arrangements of specified functional modules with established boundaries between different cell types and microenvironments.